# Do Not Marginalize Mechanisms, Rather Consolidate!

**Moritz Willig**
Technical University of Darmstadt
moritz.willig@cs.tu-darmstadt.de

**Matej Zečević**
Technical University of Darmstadt
matej.zecevic@tu-darmstadt.de

**Devendra Singh Dhami**
Eindhoven University of Technology*
Hessian Center for AI (hessian.AI)
d.s.dhami@tue.nl

**Kristian Kersting**
Technical University Darmstadt
Hessian Center for AI (hessian.AI)
German Research Center for AI (DFKI)
kersting@cs.tu-darmstadt.de

## Abstract

Structural causal models (SCMs) are a powerful tool for understanding the complex causal relationships that underlie many real-world systems. As these systems grow in size, the number of variables and complexity of interactions between them does, too. Thus, becoming convoluted and difficult to analyze. This is particularly true in the context of machine learning and artificial intelligence, where an ever increasing amount of data demands for new methods to simplify and compress large scale SCM. While methods for marginalizing and abstracting SCM already exist today, they may destroy the causality of the marginalized model. To alleviate this, we introduce the concept of *consolidating causal mechanisms* to transform large-scale SCM while preserving consistent interventional behaviour. We show consolidation is a powerful method for simplifying SCM, discuss reduction of computational complexity and give a perspective on generalizing abilities of consolidated SCM.

## 1 Introduction

Even complex real world systems might be modeled using structural causal models (*SCM*) [Pearl, 2009] and several methods exist for doing so automatically from data [Spirtes et al., 2000, Pearl, 2009, Peters et al., 2017]. While technically reflecting the causal structure of the systems under consideration, SCM might not entail intuitive interpretations to the user. Large scale SCM like, appearing for example in genomics, medical data [Squires et al., 2022, Ribeiro-Dantas et al., 2023] or machine learning [Schölkopf et al., 2021, Berrevoets et al., 2023], may become increasingly complex and thereby less interpretable. Contrary to this, computing average treatment effects might be too uninformative given the specific application, as the complete causal mechanism is compressed into a single number. Ideally a user could express the factors of interest and yield a reduced causal system that isolates the relevant mechanism from the rest of the model.

In contrast to other probabilistic models, SCM model the additional aspect of *interventions*. Consider for example a row of dominoes and its corresponding causal graph as shown in Figure 1. If the starting stone it tipped over, it will affect the following stones, causing the whole row to fall. Humans usually have a good intuition about predicting the unfolding of such physical systems [Gerstenberg, 2022, Beck and Riggs, 2014, Zhou et al., 2023]. Second to that, it is easy to imagine what would happen, if we were to hold onto a domino stone, that is, intervening actively upon the domino sequence. Alternatively, we can programmatically simulate these systems to reason about their outcomes. A simulator tediously computes and updates positions, rotations and collision states of all objects in

---

*DSD contributed while being with hessian.AI and TU Darmstadt before joining TU\e.

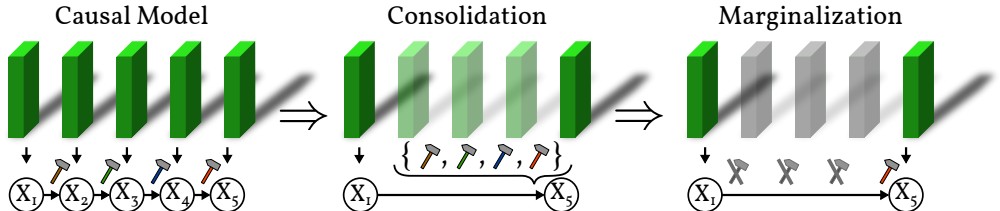

Figure 1: **Consolidation vs. Marginalization.** Even simple real-world systems, like this row of dominoes, are composed of numerous intermediate steps. Classical structural causal models require the explicit evaluation of the individual structural equations to respect possible interventions along the computational chain and yield the final value of $X_5$. The intermediate steps $(X_2, X_3, X_4)$ might be *marginalized* to obtain a simplified functional representation. Marginalization, however, loses some causal interpretation of the process, as interventions on the marginalized variables can no longer be performed. *Consolidation* of causal models simplifies the graph structure (compare to Appendix D.1), while respecting interventions on the marginalized variables. Thus, preserving the ability to intervene on the underlying causal mechanisms. (Best viewed in color.)

the system. Depending on the abstraction level of our SCM, computations might be simplified to represent individual stones as binary variables, indicating a stone standing up or getting pushed over. Nonetheless, classical evaluation of simplified SCM is still performed step by step to be able to respect possible interventions on the individual stones. Given that we might only be interested in the outcome. That is, whether or not the last stone will tip over, computing all intermediate steps seems to be a waste of computation, as already noted by Peters and Halpern [2021]. Under these premises, we are interested in preserving the ability to intervene while also being computationally efficient. Classical marginalization [Pearl, 2009, Rubenstein et al., 2017] is of no help to us as it destroys the causal aspect of interventions attached to the variables.

**Consolidation vs. Marginalization.** By marginalizing we do not only remove variables, but also all associated interventions, destroying the causal mechanisms of the marginalized variables. The insight of this paper, as alluded to in Figure 1 (center), is that there exists an intermediate tier of *consolidated* models that fill the gap between unaltered SCM and ones with 'classical' marginalization applied. Consolidation simplifies the causal graph by compressing computations of consolidated variables into the equations of *compositional variables* that are functionally equivalent to the initial model, while still respecting the causal effects of possible interventions. As such consolidation generalizes marginalization in the sense that marginalization can be modeled by consolidating without interventions ($\mathcal{I} = \emptyset$; see Def. 1 and Sec. 3). If questions involve causality, then consolidation necessarily needs to be considered since it can actually handle interventions (all cases where $\mathcal{I} \neq \emptyset$). If causal questions are not of concern, then marginalization can be considered as remaining the 'standard' marginalization procedure. One perspective on our approach is to describe consolidation as 'intervention preserving marginalization'.

**Structure and Contributions of this paper.** In section two we discuss the foundation of SCM and related work. In section three we formally establish the composition of structural causal equations, partitioned SCM, and finally consolidated SCM. In section four we discuss the possible compression and computational simplifications resulting from consolidating models. We present an applied example for simplifying time series data and in a second example demonstrate how consolidation reveals the policy of a game agent. Finally, in section five, we discuss generalizing abilities of our method and provide a perspective on the broader impact and further research. The technical contributions of this paper are as follows:

- We define *causal compositional variables* that yield functionally equivalent distributions to SCM under intervention.

- We formalize *(partially) consolidated SCM* by partitioning SCM under a constraint that guarantees the consistent evaluation with respect to the initial SCM.

- We discuss conditions under which consolidation leads to compressed causal equations.

- We demonstrate consolidation on two examples. First, obtaining a causal model of reduced size and, secondly, revealing the underlying policy of a causal decision making process.

## 2 Preliminaries and Related Work

In general we write sets of variables in bold upper-case ($\mathbf{X}$) and their values in lower-case ($\mathbf{x}$). Single variables and their values are written in normal style ($X, x$). Specific elements of a set are indicated by a subscript index ($X_i$). Probability distributions over a variable $X$ or a set of variables $\mathbf{X}$ are denoted by $\mathrm{P}_X$ and $\mathrm{P}_{\mathbf{X}}$ respectively. A detailed list of notation can be found in Appendix E.

Structural Causal Models provide a framework to formalize a notion of causality via graphical models [Pearl, 2009]. From a computational perspective, structural equation models (SEM) can be considered instead of SCM [Halpern, 2000, Spirtes et al., 2000]. While focusing on computational aspects of consolidating causal equations, we use Pearl's formalism of SCM. Modeling causal systems using SCM over SEM does not affect our freedom, as Rubenstein et al. [2017] show consistency between both frameworks. Similar to earlier works of Halpern [2000], Beckers and Halpern [2019] and Rubenstein et al. [2017], we do not assume independence of exogenous variables and model SCM with an explicit set of *allowed interventions*.

**Definition 1** *A structural causal model is a tuple* $\mathcal{M} = (\mathbf{V}, \mathbf{U}, \mathbf{F}, \mathcal{I}, \mathrm{P}_{\mathbf{U}})$ *forming a directed acyclic graph* $\mathcal{G}$ *over variables* $\mathbf{X} = \{X_1, \ldots, X_K\}$ *taking values in* $\boldsymbol{\mathcal{X}} = \prod_{k \in \{1 \ldots K\}} \mathcal{X}_k$ *subject to a strict partial order* $<_{\mathbf{X}}$*, where*

- $\mathbf{V} = \{X_1, \ldots, X_N\} \subseteq \mathbf{X}, N \leq K$ *is the set of endogenous variables.*

- $\mathbf{U} = \mathbf{X} \setminus \mathbf{V} = \{X_{N+1}, \ldots, X_K\}$ *is the set of exogenous variables.*

- $\mathbf{F}$ *is the set of deterministic structural equations,* $V_i := f_i(\mathbf{X}')$*, where the parents are* $\mathbf{X}' \subseteq \{X_j \in \mathbf{X} \,|\, X_j <_{\mathbf{X}} V_i\}$*.*

- $\mathcal{I} \subseteq \{\{I_{i,v_i}\}_{i \subseteq \{1 \ldots N\}}\}_{\mathbf{v} \in \boldsymbol{\mathcal{X}}}$ *where* $v_i$ *is the i-th element of* $\mathbf{v}$*,* $I_{i,v_i}$ *indicates an intervention* $do(X_i = v_i)$ *and such that* $\mathbf{J} \subset \mathbf{I} \in \mathcal{I} \to \mathbf{J} \in \mathcal{I}$*.* $\mathbf{I}$ *is the set of perfect interventions under consideration. A perfect intervention* $do(V_i = v_i)$ *replaces the unintervened* $f_i$ *by the constant assignment* $V_i := v_i$*.*

- $\mathrm{P}_{\mathbf{U}}$ *is the probability distribution over* $\mathbf{U}$*.*

As we focus on computational aspects of SCM, we do not regard exogenous variables to be latent, but rather consider them to take values which are not under control of the causal system itself. As such, their values are not determined via any structural equation. By construction of $\mathcal{I}$ at most one intervention on any specific variable can be included in any intervention set $\mathbf{I}$. The additional constraint enforces that $\mathcal{I}$ is closed under subsets, i.e. that any subset of any $\mathbf{I} \in \mathcal{I}$ is also part of $\mathcal{I}$. This condition is placed to yield valid intervention sets when partitioning the SCM. Every $\mathcal{M}$ entails a DAG structure $\mathcal{G} = (\mathbf{X}, \mathcal{E})$ consisting of vertices $\mathbf{X}$ and edges $\mathcal{E}$, where a directed edge from $X_j$ to $X_i$ exists if $\exists x_0, x_1 \in \mathcal{X}_j . f_i(\mathbf{x}', x_0) \neq f_i(\mathbf{x}', x_1)$. For every variable $X_i$ we define $\mathrm{ch}(X_i), \mathrm{pa}(X_i)$ and $\mathrm{an}(X_i)$ as the set of direct children, direct parents and ancestors respectively, according to $\mathcal{G}$.[2] Additionally, every $\mathcal{M}$ entails an observational distribution $\mathrm{P}_{\mathcal{M}}$[3] by propagating $\mathrm{P}_{\mathbf{U}}$ through the structural equations. Any perfect intervention $I$ on a variable $X_i$ replaces $f_i$ with a new probability distribution $\mathrm{P}_I$. As a consequence $\mathcal{M}$ entails infinitely many intervened distributions $\mathrm{P}_{\mathcal{M}}^{\mathbf{I}}$.

**Related Work.** Several works acknowledge the need for model simplification when working with causal models at different levels of modeling detail or finding consistent mappings between two already existing causal models [Rubenstein et al., 2017, Chalupka et al., 2016, Beckers et al., 2020, Zennaro et al., 2023, Brehmer et al., 2022]. However, whenever providing explicit methods of mapping SCM, marginalization is considered as a tool of removing variables. Several other works have been dedicated to proving consistency and identifiability results for grouping or clustering variables in general [Anand et al., 2022, Squires et al., 2022]. Works on $\tau$ abstractions by [Beckers and Halpern, 2019, Beckers et al., 2020] focus on simplifying models by mapping between SEM of different levels of abstractions. With regard to computational aspects, Rubenstein et al. [2017]

---

[2] We define $\mathrm{ch}(\mathbf{X}), \mathrm{pa}(\mathbf{X})$ and $\mathrm{an}(\mathbf{X})$ for sets of variables $\mathbf{X}$, as the union of sets gained by individual variable evaluations, e.g., $\mathrm{pa}(\mathbf{X}) = \bigcup_{X \in \mathbf{X}} \mathrm{pa}(X)$.

[3] We always reference a distribution with respect to some SCM $\mathcal{M}$, therefore, if we write $\mathrm{P}_{\mathcal{M}}$ then this the distribution over the full variable set, that is, $\mathrm{P}_{\mathbf{X}}$.

demonstrate the causal consistency of SEM, providing simplifications results for marginalizing SEM. However, their theorems (cf. Sec.5) explicitly exclude interventions on the marginalized variables.

# 3 Consolidation of Causal Graphical Structures

In this section, we present an approach to consolidating structural equation systems under intervention. This is the key contribution of this work compared to previous works that only considered marginalization of unintervened subsystems [Pearl, 2009, Peters et al., 2017, Rubenstein et al., 2017]. The focus is on computational aspects of marginalizing intermediate variables while preserving effects of interventions. A formalization of marginalizing intervenable structural equation systems is introduced in this section. Section 4 examines conditions under which consolidation leads to an actual reduction in complexity, followed by two practical examples.

We start with the definition of a *Causal Compositional Variable* (CCV) that has similar semantics to cluster DAGs [Anand et al., 2022], in that both capture the causal semantics over a set of variables. In contrast to cluster DAGs, CCVs are defined over an SCM $\mathcal{M}$ and moreover expose an interface for explicitly applying interventions to the individual variables inside the CCV. We define a CCV with a corresponding function $\rho$, that takes the exogenous variables $\mathbf{U}$ as its input and outputs the values of a subset $\mathbf{E} \subseteq \mathbf{V}$. Thus we write $\rho_{\mathbf{E}}$ to denote the set of computed variables. To be able to condition on interventions, $\rho$ takes the set of interventions $\mathbf{I}$ as it would be applied to the SCM as its second argument.

**Definition 2 (Causal Compositional Variable)** *A variable $X_{\mathbf{E}}^{\mathbf{I}} := \rho_{\mathbf{E}}(\mathbf{U}, \mathbf{I}) \in \mathcal{X}^{|\mathbf{E}|}$* [4] *is a causal compositional variable over some subset $\mathbf{E} \subseteq \mathbf{V}$ of an SCM $\mathcal{M}$, if a consolidation function $\rho_{\mathbf{E}} : (\mathcal{X}^{|\mathbf{U}|}, \mathcal{I}) \to \mathcal{X}^{|\mathbf{E}|}$ exists for which $\mathrm{P}_{X_{\mathbf{E}}^{\mathbf{I}}} = \mathrm{P}_{\mathbf{E}}^{\mathbf{I}}$ for all $\mathbf{I} \in \mathcal{I}$, where $\mathrm{P}_{\mathbf{E}}^{\mathbf{I}}$ is the distribution of target variables $\mathbf{E}$ in under some intervention set $\mathbf{I}$.*

Put in simple terms, $\rho_{\mathbf{E}}$ yields the same values for $\mathbf{E}$ as would be determined by evaluation of the initial SCM $\mathcal{M}$ given any $\mathbf{u} \sim \mathrm{P}_{\mathbf{U}}$. Naturally, there always exists such a function $\rho_{\mathbf{E}}$ for every $\mathbf{E} \subseteq \mathbf{V}$, which is computing $\mathbf{e} \in \mathbf{E}$ via evaluation of $\mathcal{M}$ itself. However, $\rho_{\mathbf{E}}$ is not required to adhere to the computation sequence imposed by the structural causal model $\mathcal{M}$. In particular, $\rho_{\mathbf{E}}$ is not required to explicitly compute the intermediate values of any $V_i \in \mathbf{V} \setminus \mathbf{E}$, which gives way to simplifying internal computations. As such a CCV serves as a possible stand-in for replacing whole SCM by a function of possibly simpler computational complexity:

**Definition 3 (Consolidated SCM)** *Given a causal compositional variable $X_{\mathbf{E}}^{\mathbf{I}} := \rho_{\mathbf{E}}(\mathbf{U}, \mathbf{I})$ and some base SCM $\mathcal{M}$, we call $\mathcal{M}_{\mathbf{E}} = (\mathbf{E}, \mathbf{U}_{\mathcal{M}}, \rho_{\mathbf{E}}, \mathcal{I}_{\mathcal{M}}, \mathrm{P}_{\mathbf{U}_{\mathcal{M}}})$ a consolidated SCM.*

The distributions of the consolidated SCM $\mathrm{P}_{\mathcal{M}_{\mathbf{E}}}$ are not equal to that of the initial SCM $\mathrm{P}_{\mathcal{M}}$, since $\mathcal{M}_{\mathbf{E}}$ only computes a subset $\mathbf{E} \subseteq \mathbf{V}$ of all endogenous variables. However, for that subset $\mathbf{E}$, the initial SCM and consolidated model yield the same $\mathrm{P}_{\mathbf{E}}^{\mathbf{I}}$ for all $\mathbf{I} \in \mathcal{I}$.

## 3.1 Partition of Structural Causal Models

So far, we considered constructing compositional variables from SCM such that they exhibit functional equivalent behaviour and, by doing so, are able to replace base SCM by consolidated SCMs using CCVs. However, compositional variables trade off the 'semantic' graph structure of a classical SCM against a computationally simpler (refer to Sec. 4), but 'black box' function. In practice we might, therefore, only want to replace certain parts of an SCM with consolidated functions. To achieve this goal, we formalize a partition of base SCM into multiple *sub SCM*. Multiple other works have considered the existence of joint variable clusters within SCM [Anand et al., 2022, Squires et al., 2022]. However, allowing for arbitrary clusters may induce cycles to the model, which would be undesirable. In our work we constrain the clustering by requiring partitions that enforce acyclicity and, therefore, ensure a well defined evaluation order that is consistent with that of the initial SCM.

Endogenous nodes of a base SCM $\mathcal{M}$ can be partitioned into $L$ mutually exclusive exhaustive components $\mathcal{A} = \{\mathbf{A}_i \in \mathcal{P}(\mathbf{V}) \setminus \emptyset : i \in \{1, \ldots, L\}\}$ with $\forall \mathbf{A}_i, \mathbf{A}_j \in \mathcal{A} : i \neq j \Rightarrow \mathbf{A}_i \cap \mathbf{A}_j = \emptyset$ and $\bigcup_{i \in \{1 \ldots L\}} \mathbf{A}_i = \mathbf{V}$. We also call $\mathcal{A}$ the (exhaustive) partition. We can use any cluster $\mathbf{A} \in \mathcal{A}$ to

---

[4]To be precise $\mathcal{X}^{|\mathbf{E}|} = \prod_{V_i \in \mathbf{E}} \mathcal{X}_i$, where $\prod$ is the n-ary Cartesian product.

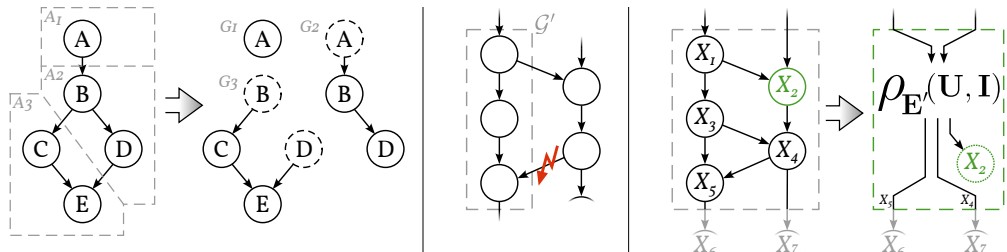

Figure 2: **Consolidating SCMs.** (**Left**) The base graph of an exemplary SCM $\mathcal{M}$ gets deconstructed into three sub SCM using the partition set $\mathcal{A} = \{\{A\}, \{B, D\}, \{C, E\}\}$. Exogenous variables are displayed with dashed circles. (**Center**) A subgraph $\mathcal{G}'$ within a larger base SCM. There exists a directed path that exits and re-enters $\mathcal{G}'$, thus preventing self-enclosed evaluation of $\mathcal{G}'$. (**Right**) Consolidation of a sub SCM into a multivariate compositional variable. $X_2$ is an aspect variable chosen by the user, ($X_2 \in \mathbf{E}$). $X_4$ and $X_5$ are needed for further computation, thus $\mathbf{E}' = \{X_2, X_4, X_5\}$. The value of $X_2$ is computed via $\rho_{\mathbf{E}'}$ and interventions can be performed via its parameter $\mathbf{I}$. The dotted line indicates that $X_2$ is not an 'independent' variable. Specifically it is not allowed to intervene on $X_2$ via 'edge cutting', making it independent of $\rho_{\mathbf{E}'}$ (and in consequence causing $\rho_{\mathbf{E}'}$ to compute inconsistent values for $X_6$ and $X_7$).

form a new sub SCM $\mathcal{M}_{\mathbf{A}}$: $\mathcal{M}_{\mathbf{A}} = (\mathbf{A}, \mathbf{U}_{\mathbf{A}}, \mathbf{F}_{\mathbf{A}}, \mathcal{I}_{\mathbf{A}}, \mathrm{P}_{\mathbf{U}_{\mathbf{A}}})$ where $\mathbf{U}_{\mathbf{A}} = \mathrm{pa}(\mathbf{A}) \setminus \mathbf{A}$, $\mathbf{F}_{\mathbf{A}}$ are the structural equations of $\mathbf{A}$, and $\mathrm{P}_{\mathbf{U}_{\mathbf{A}}}$ is the distribution over $\mathbf{U}_{\mathbf{A}}$ induced by the base SCM. As some intervention $\mathbf{I} \in \mathcal{I}$ might intervene on variables which are no longer part of $\mathcal{M}_{\mathbf{A}}$, we define a mapping $\psi_{\mathbf{A}} : \mathbf{I} \to \mathbf{I}_{\mathbf{A}}$ which removes those invalid interventions: $\psi_{\mathbf{A}}(\mathbf{I}) := \{do(V_i = v_i) \in \mathbf{I} : V_i \in \mathbf{V}_{\mathbf{A}}\}$. Consequently we define $\mathcal{I}_{\mathbf{A}} := \{\psi_{\mathbf{A}}(\mathbf{I}) : \mathbf{I} \in \mathcal{I}\}$. As expected, whenever a set of interventions $\mathbf{I}$ does not intervene on any $V \in \mathbf{A}$, $\psi_{\mathbf{A}}$ maps it to the empty set. For notational brevity, we assume the implicit application of $\psi$ on any $\mathbf{I}$ whenever we apply interventions to a sub SCM. Figure 2 (left) presents an exemplary construction of sub SCM from a given partition of a base SCM.

Unconstrained partitions may divide SCM in an arbitrary way. To guarantee an evaluation order of the individual sub SCM that is consistent with that of the base SCM we need to ensure that any particular sub SCM can be evaluated in a continuous, self-enclosed manner. That is, no intermediate evaluation of external nodes, $V \notin \mathbf{A}$, is required. Figure 2 (center) illustrates a counter-example of a non-complying partition where an intermediate external evaluation to $G'$ is required. To prevent such cases we require the partitions to yield a strict partial ordering under the following definition: the binary relation $\mathbf{A}_1 \, \mathrm{R}_{\mathbf{X}} \, \mathbf{A}_2 \iff \exists A_i \in \mathbf{A}_1, A_j \in \mathbf{A}_2 : A_i <_{\mathbf{X}} A_j$ holds if at least one variable in $\mathbf{A}_1$ needs to be evaluated before some other variable in $\mathbf{A}_2$ according to $<_{\mathbf{X}}$ of the base SCM. We call a partition $\mathcal{A}$ "according to $\mathcal{M}$" iff $\mathrm{R}_{\mathbf{X}}$ is a strict partial order[5] over all $\mathbf{A} \in \mathcal{A}$.

**Definition 4 (Partitioned SCM)** *Given an exhaustive partition $\mathcal{A}$, a partitioned SCM $\mathcal{M}_{\mathcal{A}}$ for some base SCM $\mathcal{M}$ is defined as $\mathcal{M}_{\mathcal{A}} = (\bigcup \mathbf{A}_i, \bigcup \mathbf{U}_{\mathbf{A}_i}, \bigcup \mathbf{F}_{\mathbf{A}_i}, \bigcup \mathcal{I}_{\mathbf{A}_i}, \bigcup \mathrm{P}_{\mathbf{U}_{\mathbf{A}_i}}), i \in \{1 \ldots L\}$ s.t. there exists a strict partial order $\mathrm{R}_{\mathbf{X}}$ over all $\mathbf{A}_i \in \mathcal{A}$ according to $\mathcal{M}$ and every $\mathcal{M}_{\mathbf{A}_i} = (\mathbf{A}_i, \mathbf{U}_{\mathbf{A}_i}, \mathbf{F}_{\mathbf{A}_i}, \mathcal{I}_{\mathbf{A}_i}, \mathrm{P}_{\mathbf{U}_{\mathbf{A}_i}})$ forms a valid sub SCM.*

**Consistency of partitioned SCM evaluation.** To ensure for the consistent evaluation of all sub SCM $\mathcal{M}_{\mathbf{A}}$ within a partitioned SCM $\mathcal{M}_{\mathcal{A}}$ we need to ensure that the evaluation is carried out according to some $\mathrm{R}_{\mathbf{X}}$ that is compliant according to the base SCM $\mathcal{M}$.[6] Doing so, guarantees that the value of every exogenous variable $\mathbf{U}_i$ of a sub SCM $\mathcal{M}_{\mathbf{A}_s}$ – that is not truly exogenous ($\mathbf{U}_i \notin \mathcal{M}_{\mathbf{U}}$) – is computed as an endogenous variable $V_j$ inside another $\mathcal{M}_{\mathbf{A}_t}$, that is evaluated before $\mathcal{M}_{\mathbf{A}_s}$ with $\mathbf{U}_i := V_j$. For example $G_2$ in Fig. 2 (left) computes the values of $B$ and $D$, required as exogenous variables by $G_3$. Lastly, during evaluation, all $\mathcal{M}_{\mathbf{A}}$ need to agree on the same set of applied interventions. This is done by fixing a particular $\mathbf{I}'$ during evaluation and computing the intervention set $\mathbf{I}'_{\mathbf{A}} := \psi_{\mathbf{A}}(\mathbf{I}')$ specific to every $\mathcal{M}_{\mathbf{A}}$. An algorithm for evaluating partitioned SCM and its proof of consistency are presented in Appendix A.

---

[5]In particular $\mathrm{R}_{\mathbf{X}}$ is a strict partial order, if it is asymmetric: $\forall \mathbf{A}_1, \mathbf{A}_2 \in \mathcal{A} : \mathbf{A}_1 \, \mathrm{R}_{\mathbf{X}} \, \mathbf{A}_2 \Rightarrow \neg(\mathbf{A}_2 \, \mathrm{R}_{\mathbf{X}} \, \mathbf{A}_1)$, implying that the evaluation of no two sub SCM mutually depend on each other.

[6]As $<_{\mathbf{X}}$ is a partial order, there may exist multiple total orders which comply with the partial ordering of $\mathcal{M}$.

---

**Algorithm 1** Consolidation of Structural Causal Models

---

1: **procedure** CONSOLIDATE($\mathcal{M}, \mathcal{A}, \mathbf{E}$)
2:     **for all** $\mathbf{A}_i$ in $\mathcal{A}$ **do**
3:         $\mathbf{E}_i \leftarrow \mathbf{A}_i \cap \mathbf{E}$                                  $\triangleright$ Filter aspect variables for the current $\mathbf{A}_i$.
4:         $\mathbf{E}'_i \leftarrow \mathbf{E}_i \cup (\mathrm{pa}(\mathbf{V} \setminus \mathbf{A}_i) \cap \mathbf{A}_i)$     $\triangleright$ Add variables that are required by other sub SCM.
5:         $\mathbf{U}_{\mathbf{A}_i} \leftarrow \mathrm{pa}(\mathbf{A}_i) \setminus \mathbf{A}_i$                   $\triangleright$ Define exogenous variables and interventions.
6:         $\mathcal{I}_{\mathbf{A}_i} \leftarrow \{\psi_{\mathbf{A}_i}(\mathbf{I}) : \mathbf{I} \in \mathcal{I}\} = \{\{do(X_i = v) \in \mathbf{I} : X_i \in \mathbf{A}_i\} : \mathbf{I} \in \mathcal{I}\}$
7:         $\rho_{\mathbf{E}'_i}(\mathbf{U}_{\mathbf{A}_i}, \mathbf{I}) \leftarrow \{\mathbf{F}_j : X_j \in \mathbf{A}_i\}$     $\triangleright$ Define a causal compositional variable via $\rho_{\mathbf{E}'_i}$.
8:         $\rho^{\star}_{\mathbf{E}'_i} \leftarrow \mathrm{argmin}_{\rho'_{\mathbf{E}'_i}} \mathcal{K}(\rho'_{\mathbf{E}'_i})$                 $\triangleright$ Minimize representation (see Sec. 4).
            s.t. $\rho'_{\mathbf{E}'_i}(\mathbf{U}_{\mathbf{A}_i}) = \rho_{\mathbf{E}'_i}(\mathbf{U}_{\mathbf{A}_i})$
9:         $\mathcal{M}_{\mathbf{A}_i, \mathbf{E}} \leftarrow (\mathbf{E}'_i, \mathbf{U}_{\mathbf{A}_i}, \rho^{\star}_{\mathbf{E}'_i}, \mathcal{I}_{\mathbf{A}_i}, \mathrm{P}_{\mathbf{U}_{\mathbf{A}_i}})$ $\triangleright$ Define the sub SCM resulting from $\mathbf{A}_i$ and $\mathbf{E}$.
10:     **end for**
11:     $\mathcal{M}_{\mathcal{A}, \mathbf{E}} \leftarrow (\bigcup \mathbf{E}'_i, \bigcup \mathbf{U}_{\mathbf{A}_i}, \bigcup \rho^{\star}_{\mathbf{E}'_i}, \bigcup \mathcal{I}_{\mathbf{A}_i}, \bigcup \mathrm{P}_{\mathbf{U}_{\mathbf{A}_i}}), i \in \{1 \ldots |\mathcal{A}|\}$     $\triangleright$ Merge all $\mathcal{M}_{\mathbf{A}_i, \mathbf{E}}$.
12:     **return** $\mathcal{M}_{\mathcal{A}, \mathbf{E}}$                                     $\triangleright$ Return the consolidated SCM.
13: **end procedure**

---

Figure 3: **CONSOLIDATE Algorithm.** The above pseudo-code summarizes the consolidation algorithm as described in this paper by utilizing causal compositional variables and partitioned SCM to obtain simplified SCM. Depending on the use-case Step 11 might be skipped and the partitioned SCM might be returned instead.

**Partial consolidation of SCM.** Having defined partitioned SCM allows us to selectively swap out arbitrary sub SCM by their consolidated SCM. In Def. 3 we placed no constraints on $\mathbf{E}$ to allow for arbitrary consolidation of variables. For sub SCM $\mathcal{M}_{\mathbf{A}}$ that appear within a partitioned SCM $\mathcal{M}_{\mathcal{A}}$ we need to constrain $\mathbf{E}$ to additionally include all variables $V \in \mathbf{U}_{\mathbf{A}}$ such that evaluation of $\mathcal{M}_{\mathbf{A}}$ additionally computes all variables needed as exogenous by other sub SCMs. Fig. 2 (right) shows an exemplary sub SCM with $X_2$ (green) chosen as a relevant aspect variable by the user, and $X_4$, $X_5$ being required by evaluations of subsequent SCM. Thus $\mathbf{E}' = \{X_2, X_4, X_5\}$. Whether to consider $\mathbf{E}$ or $\mathbf{E}'$ depends on the standpoint of the user. From a computational perspective $\mathbf{E}'$ is important as it holds all variables that need to be computed by $\rho$. On the other hand, the set $\mathbf{E}$ captures aspects of the SCM important to the user i.e., variables of interest. We will therefore refer to sub SCM with $\mathcal{M}_{\mathbf{E}'}$ (and in the same breath write $\rho_{\mathbf{E}'}$) but use $\mathcal{M}_{\mathcal{A}, \mathbf{E}}$ (see the following Def. 5) to retain the initial set of variables chosen by the user. Having defined consolidated SCM $\mathcal{M}_{\mathbf{E}}$, partitioned SCM $\mathcal{M}_{\mathcal{A}}$ and the required constraint on $\mathbf{E}$ we are now equipped with the tools to define a partially consolidated SCM that yields a consistent $\mathrm{P}_{\mathbf{E}}$ with the base SCM.

**Definition 5 (Partially Consolidated SCM)** *A partially consolidated SCM $\mathcal{M}_{\mathcal{A}, \mathbf{E}}$ is a partitioned SCM $\mathcal{M}_{\mathcal{A}}$ such that a subset of sub SCM $\mathcal{M}_{\mathbf{A}}$ are being replaced by consolidated SCM $\mathcal{M}_{\mathbf{E}'}$ where* $\mathbf{E}'_i := \{V_i \in \mathbf{V}_{\mathbf{A}} : (V_i \in \mathbf{E}) \vee (\exists \mathcal{M}_{\mathbf{A}'} = (\mathbf{V}', \mathbf{U}', \mathbf{F}', \mathcal{I}', \mathrm{P}'_{\mathbf{U}'}).V_i \in \mathbf{U}')\}$.

Algorithm 1 summarizes all considerations of this chapter, starting out from a subset $\mathbf{E}$ and partition $\mathcal{A}$ up to a (partially) consolidated SCM $\mathcal{M}_{\mathcal{A}, \mathbf{E}}$. An exemplary step-by-step application of the algorithm can be found in Appendix D.3. The purpose of the argmin operation in Line 8 is to minimize complexity of $\rho'_{\mathbf{E}'_i}$ by finding a minimal encoding. We discuss this step in more detail in the following section. After formally introducing consolidation, we are ready to illustrate its applicability.

## 4 Compression of Causal Equations

Model consolidation can lead to compression by reducing the model's graph structure and leveraging redundant computations across equations. This may result in smaller, simpler models that are computationally more efficient and easier to analyze. Compressing structural equations to a minimal representation is highly dependent on the equations under consideration and probably incomputable for most problems. As there is ultimately no way of measuring compressibility of SCM by only considering their connecting graph structure, we provide a discussion with regard to some of the

information-theoretical implications. Specifically, we discuss compression properties for some of the basic structures appearing within SCM; namely chains, forks and colliders. In this section, we, first, analyze how consolidated models may leverage redundant computations for reducing complexity within chained equation in general. Second, we give a condition under which equations, and their interventions can be dropped from the consolidation model altogether. Thirdly, we analyse how interventions within the consolidated model affect our ability to compress equations. Lastly, we will walk through two examples of model compression.

**General compression of equation systems.** Using our formalization of (partially) consolidated SCM, we now have the chance to replace certain parts of an SCM with computationally simpler expressions. The notion of what a 'simple' expression may be, varies depending on the application and is subjective to the user. To define a measurable metric, we reside to a simplified notion of complexity by measuring the representation length of our consolidated equations. We assume that all structural equations of an SCM can be expressed in terms of elementary operators, where each term contributes the same amount of complexity. As such, we can apply Kolmogorov complexity $\mathcal{K}$ [Kolmogorov, 1963]. Then a desirable minimal representation of a structural equation $f_i^\star$ is one that minimizes $\mathcal{K}(f_i)$: $f_i^\star := \operatorname{argmin}_{f_i'} \mathcal{K}(f_i')$ s.t. $f_i'(\operatorname{pa}(X_i)) = f_i(\operatorname{pa}(X_i))$.

Classical marginalization reduces the number of variables in a graph. To keep the model consistent after marginalization, all children $\mathbf{B} := \operatorname{ch}(A)$ of a marginalized variable $A$ additionally need to incorporate the values of $\operatorname{pa}(A)$ to accommodate for the causal effects that where previously flowing through $A$ into $\mathbf{B}$. This modifies the structural equations of any $B \in \mathbf{B}$, $f_B' := f_B \circ f_A$, where $f_B$ and $f_B'$ are the structural equations of $B$ before and after marginalization, respectively. Evaluation of the separate equations $f_A, f_B$ provides an upper bound on the complexity of the composed representation $\mathcal{K}(f_B'^\star) \leq \mathcal{K}(f_A^\star) + \mathcal{K}(f_B^\star)$ [Zvonkin and Levin, 1970]. Since the consolidated system is not required to compute $A$ explicitly, the encoding length of $f_B'^\star$ might resort to directly computing $B$ from the values of $\operatorname{pa}(A)$. Also, the chain rule for Kolmogorov complexity only considers the case of reproducing $f_A$ and $f_B$ in their initial forms. In addition to that, we might also use semantic rules to reduce equation length, e.g. by collapsing consecutive additions $\forall a, b \in \mathbb{R}. \exists c \in \mathbb{R}. a + b = c$ and so on. Whether consolidation actually leads to simplified equations depends strongly on the specific equations and their connecting graph structure. No simplification effects occur in cases of already minimal systems, while strong cancellation occurs in the case of $f_B, f_A$ being inverses to each other (see Appendix B.1). Lastly, we want to refer to Appendix B.2, where we showcase the insufficiency of matrix composition to obtain minimal function representations in the case of linear systems.

**Marginalizing child-less variables.** Regardless of the particular causal graph structure, all equations which do not affect $P_{\mathbf{E}'}$ can be removed from the model to reduce its overall complexity. In particular we point out that $P_{\mathbf{E}'}$ is invariant to all $X \notin \operatorname{an}(\mathbf{E}')$. By the following deduction we infer that we can always consolidate all child-less variables (if not part of $\mathbf{E}'$ themselves) from $\mathcal{M}$: $\forall X \in \mathbf{X} \setminus \mathbf{E}'.[(\operatorname{ch}(X) = \emptyset) \Rightarrow (\forall X' \in \mathbf{X}. X \notin \operatorname{pa}(X')) \Rightarrow (\forall X' \in \mathbf{X}. X \notin \operatorname{an}(X')) \Rightarrow X \notin \operatorname{an}(\mathbf{E}')]$. Since child-less variables do not affect $P_{\mathbf{E}'}$, we can not only consolidate but *marginalize* them. (Reducing to the same scenario as in Rubenstein et al. [2017, Thm. 9]). Therefore, we are allowed to drop interventions $do(X_i = c)$ with $X_i \notin \operatorname{an}(\mathbf{E}')$ from the set of allowed interventions. This process can be applied repeatedly until we have pruned the SCM from all child-less variables irrelevant to $\mathbf{E}'$.

### 4.1 Simplifying Graphical Structures

In contrast to marginalization, consolidation preserves the effects of interventions for consolidated variables. This effectively adds conditional branching to every structural equation $f_i$ if some $\mathbf{I} \in \mathcal{I}$ with $do(V_i = c) \in \mathbf{I}$ exists:

$$V_i := \begin{cases} c & \text{if } do(V_i = c) \in \mathbf{I} \\ f_i(\operatorname{pa}(V_i)) & \text{else} \end{cases} \tag{1}$$

While conditional branching might prevent us from compressing equations, we consider that not all variables might be affected by interventions. As such, we might be able to utilize local structures within the graph to simplify equations. In the following we briefly discuss the possibilities of simplifying chains, forks and collider structures within the graphs of SCMs:

**Simplifying Chains.** Consolidating chains of consequent variables corresponds to 'stacking' structural equations and computing the last non-consolidated variable directly. In the general case,

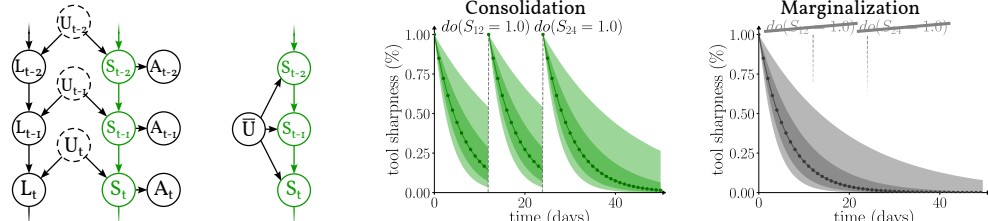

Figure 4: **Consolidating a real world mechanism.** (**Left**) The causal time-series model of a milling machine, representing tool length $L$, utilization $U$, sharpness $S$ and accuracy $A$. (**Center-Left**) Removing child-less nodes $L_t$ and $A_t$ and controlling for the parents $U_t$ yields a simplified causal structure. (**Center-Right**) Plots for the consolidated structural equation of $S$. Colored areas show the effects of varying $\bar{U}$ by one and two sigma ($\pm 0.05$, $\pm 0.1$) respectively. Dashed grey lines indicate interventions, which are respected truthfully by the consolidated function. (**Right**) Marginalization, likewise, simplifies the model, but does not allow us to investigate the effects of interventions.

conditional branching complicates the simplification of the stacked equations into a single closed-form representation. When considering the case of marginalization, that is without considering interventions, as done in Rubenstein et al. [2017], composition of equations turns into direct function composition $X_i := f_i \circ f_{i-1} \circ f_{i-2} \circ \ldots$. To this end, a complexity bound on chained equations over finite discrete domains is discussed in Appendix B.3, as well as consolidation of the motivating dominoes example in Appendix D.1.

**Simplifying Forks.** Consolidating the parent node $B$ of a fork structure, $A \leftarrow B \rightarrow C$, might lead to a duplication of $f_B$ into the equations of both child nodes, $f'_A := f_A \circ f_B$, $f'_C := f_C \circ f_B$. If $\text{pa}(B) \subset \mathbf{U}$, then $A$ and $C$ will be confounded by exogenous variables. This is the reason why we did not require independence of exogenous variables in Def. 1. Still, consistency with the initial SCM is guaranteed, since we require all structural equations to be deterministic. As a consequence, every evaluation of the duplicated structural equations $f_B$ inside $f'_A$ and $f'_C$ yields the same value when given the same inputs. While determinism of structural equations is formally required, we illustrate a consistent reparameterization of non-deterministic models in Appendix C.

**Simplifying Colliders.** Colliders are the most promising graphical structures for simplifying equations. When consolidating $A$ and $C$ of a collider $A \rightarrow B \leftarrow C$, we might leverage mutual information between $f_A$ and $f_C$ to simplify $f_B$. Especially in the case of $\text{pa}(A) = \text{pa}(C)$, consider $A \leftarrow X \rightarrow C$ for example, we might be able to discard $f_A$ and $f_C$ altogether and compute $B$ directly from $X$.

### 4.2 Time Series Example: Tool Wear

We will now demonstrate a simple application of consolidation for a possibly more applied scenario. Imagine that we want to create a causal model of an industrial unit under continuous use, e.g. a milling machine. At the end of every work day the length $L$ and sharpness $S$ of the milling cutter are measured. From these measurements other metrics such as the cutting accuracy $A$ can be derived. Interventions on the process are performed by grinding the cutter, 'resetting' it to a certain sharpness. While every intervention grinds away some material, the weight and size changes are negligible for the considered aspect of accuracy. Throughout our recordings we might encounter multiple such interventions. From the data we fit a 'classical' SCM that models the time series on a day-to-day basis, $\mathbf{V}_{t-1} \rightarrow \mathbf{V}_t$. We observe the tool to loose some percentage of its sharpness per day depending on its utilization $U_t$. The intervention $do(S_t = 1)$ resets the sharpness to a constant value, while $do(S_t = 1, L_t = 1)$ models a tool replacement. As by Def. 1, $\mathcal{I}$ needs to include $do(L_t = 1)$, which might be a recalibration of the machine. Figure 4 (left) shows the initial causal graph of the time series model as defined by the following SCM:

$$\mathcal{M} = \begin{cases} \mathbf{U} & = \{\mathbf{U}_t = \mathcal{N}(0.5, 0.05^2)\} \\ \mathbf{V} & = \{\mathbf{L}_t, \mathbf{S}_t, \mathbf{A}_t\} \\ \mathcal{I} & = \mathcal{P}(\{do(\mathbf{S}_t = 1), do(\mathbf{L}_t = 1.0), do(\mathbf{S}_t = 1, \mathbf{L}_t = 1)\}) \\ \mathbf{F} & = \begin{cases} f_l(l, u) & := (1.0 - 0.002u)l \\ f_{\mathbf{s}_t}(s_{t-1}, u) & := (1.0 - 0.3u)s_{t-1} \\ f_a(s) & := 0.8s^2 \end{cases} \end{cases}$$

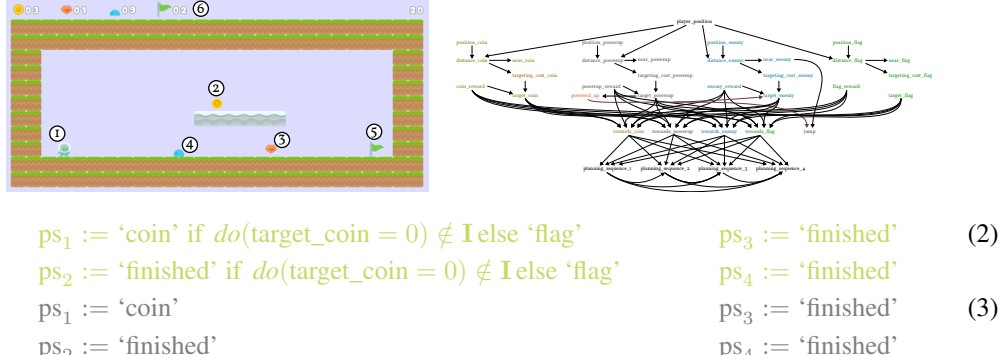

$$\text{ps}_1 := \text{'coin' if } do(\text{target\_coin} = 0) \notin \mathbf{I} \text{ else 'flag'} \qquad \text{ps}_3 := \text{'finished'} \qquad (2)$$
$$\text{ps}_2 := \text{'finished' if } do(\text{target\_coin} = 0) \notin \mathbf{I} \text{ else 'flag'} \qquad \text{ps}_4 := \text{'finished'}$$
$$\text{ps}_1 := \text{'coin'} \qquad \text{ps}_3 := \text{'finished'} \qquad (3)$$
$$\text{ps}_2 := \text{'finished'} \qquad \text{ps}_4 := \text{'finished'}$$

Figure 5: **Complex situations can be easy to understand using consolidation.** Encoding the behaviour of agents acting in game environments (**left**) often results in complex causal graphs (**right**; indeed unreadable due to complexity. A readable version is contained in the Appendix). Even very simple levels with a single agent, coin, power-up and enemy, entail causal graphs that are intuitively non-interpretable. Especially the intertwining of game mechanics and agent behaviour complicates the inference of the agents' actual policy and makes it impossible to judge its performance. In our example a suboptimal greedy policy is embedded within the causal graph, which can be made visible using consolidation (**bottom**, Eq. (2)). Please note that ps abbreviates 'planning_sequence'. In contrast to marginalization (Eq. (3)), one can still intervene on the consolidated system.

Now, we might be interested in extracting a formula for the total tool sharpness $S_t$ at an arbitrary point in time $t$. Thus, our consolidation set consists of all $S_t$, $\mathbf{E} = \{S_t\}$. Since $U_t$ is exogenous, we make the additional assumption that the utilization follows a normal distribution and we simplify to the expected value $\bar{u} = 0.5$ (Figure 4, center-left). As laid out before, we can marginalize all child-less variables $L_t$ and $A_t$ not part of $\mathbf{E'}$. As all $L_t$ are no longer part of $\mathcal{M}_{\mathbf{E'}}$, $\psi_{\mathbf{E'}}$ maps interventions $do(S_t = s_t, L_t = l_t) \to do(S_t = s_t)$. Considering the uninterved case, structural equation are now simplified via function composition, $f_{\mathbf{S}_t} := f_{\mathbf{S}_t} \circ \ldots f_{\mathbf{S}_1}$, which results in the following equation $f_{\mathbf{S}_t} := (1 - 0.3 \cdot 0.5)^t = 0.85^t$. According to Eq. (1), we now have to inspect interventions as potential candidates for conditional branching. All remaining interventions are of the form $do(S_t = 1.0)$. Applying an intervention at time $t_0$ equals shifting the following equations by the time of that last intervention $t' := t - t_0$. Finally, we arrive at the following consolidated equation:

$$f_S(t) := 0.85^{t - t_0}$$
$$\text{where} \quad t_0 = \max_i \{i \mid \exists do(S_i) \in \mathbf{I} \wedge i \leq t\}$$

Fig. 4 (center-right) shows the resulting plot of the consolidated model under interventions $do(S_{12} = 1)$ and $do(S_{24} = 1)$. We successfully demonstrated the power of consolidation models for dynamical system while preserving the ability to intervene. In theory more complex dynamical systems could be consolidated. However, as these kind of self-referential models require a more involved discussion, we kindly refer the reader to Bongers et al. [2018, 2021], Peters et al. [2022] for further considerations.

### 4.3 Revealing Agent Policy

In our second example we apply consolidation to a more complex causal graph relating the game state of a simple platformer environment to the actions of an agent. See Appendix D.4 for the full causal graph and structural equations. Throughout the level the agent ((1) in Fig. 5) can interact with a coin ((2) in Fig.), a power-up (3), an enemy (4) and the finish flag (5) to accumulate a certain reward (6) by doing so. The power-up is required to interact with the enemy. During play, the agent takes the state of the environment as its input and outputs the state of 'towards_coin', 'towards_powerup', etc. The order of the agent actions is then recorded via four 'planning_sequence_i' for $i \in \{1 \ldots 4\}$ variables. A causal graph, like the one presented in Appendix D.4, might be extracted automatically from observational data, or designed by an expert. Due to the sheer number of variables and edges, dependencies in the obtained SCM are hard to trace. To get a better understanding, we use consolidation to reveal the policy of our agent. We consolidate all endogenous variables except player-

entity 'distance' and the 'planning_sequence' variables. To be able to modify the agents behaviour, we allow interventions by forbidding the agent to target certain entities: $\mathcal{I} = \mathcal{P}(\{do(\text{target\_coin} = 0),\ do(\text{target\_enemy} = 0),\ do(\text{target\_powerup} = 0)\})$.

Like before, we consolidate equations considering the unintervened case, and then add back in conditional branching for interventions to yield equations (2) in Figure 5. Contrary to the very complex structure of the SCM, the consolidated equation reveals the actually very *simple* policy of the agent. We find from the consolidated equation that the agent only collects the coin, if not intervened upon, and then heads directly towards the flag. This insight might not be obvious from the initial SCM and, at least, is difficult to spot a priori by looking at the unconsolidated equations. When inspecting the original SCM more closely we find, that a constant factor is added to the calculation of 'targeting_cost_powerup.' This factor might serve to accommodate for the time lost when speeding up or slowing down towards a target. Furthermore, we see that the agent pursues a greedy policy, thus, never considering the overall higher reward of the power-up and enemy together. Instead the policy ignores the power-up, due to its low reward and in consequence also never targets the enemy. This behaviour not only leads to strong simplification of the SCM, but also allows us to discard the imperfect policy, without the need to run possibly costly trials, just to come to the same conclusion.

To summarize, consolidation is a strictly more powerful operation than marginalization. More examples and domains can be found in Appendix D.

## 5 Conclusions

Consolidation is a powerful tool for transforming SCMs, while preserving the causal aspect of interventions. In addition, consolidating SCMs can lead to more general models. For example, recall the tool wear example of Sec. 4.2. While the initial causal graph operates on discrete time steps our consolidated function provides a continuous relaxation of the causal process. We can evaluate it at any point in time $t \in \mathbb{R}$ and are no longer dependent on the day-to-day basis which was modeled by the initial graph. Additionally, the consolidated equation of our motivating example of rows of dominoes (compare Appendix D.1), yields a generalized formula that is independent of the actual number of dominoes that compose the row, by making use of first-order quantifiers. In our discussion of Sec. 4 we saw that we can further benefit from consolidation in all cases where the initial SCM does not already represent the smallest possible causal model. Lastly, these simplifications align with our goal of making SCMs more interpretable. Consider that the initial domino SCM only provides a 'local' view on the system, by only providing equations for every individual stone, "If stone A falls it pushes over stone B, except in the case of an intervention. If stone B falls, ..." and so on. The equation of the consolidated SCM can be directly translated into a single natural language sentence, e.g. "The last domino will fall, if the first domino is pushed over, except in the case of holding onto or pushing over a stone along the way", capturing the causal mechanisms of the system much more intuitively. Our last example of Sec. 4.3 strikingly revealed the sub-optimal, greedy agent behaviour in a game setting. While we illustrated examples that are well suited for consolidation, we are positively inclined to expect consolidation to be helpful towards a broad range of applications.

**Limitations and Broader Impact.** Throughout the paper we considered exact consolidations, in that Def. 2 requires strict equality between $\rho_{\mathbf{E}}(\mathbf{U}, \mathbf{I})$ and $\mathrm{P}_{\mathbf{E}}^{\mathbf{I}}$. This assumption might be met in logic and idealized scenarios, but may hinder consolidation of SCM in other applications due to noise inherent to the system. The definition might be relaxed by allowing for small deviations of $\rho_{\mathbf{E}}$ from the distribution of the unconsolidated SCM. Thus, relaxing the strict equality $\mathrm{P}_{X_{\mathbf{E}}^{\mathbf{I}}} = \mathrm{P}_{\mathbf{E}}^{\mathbf{I}}$ with $|\mathrm{P}_{X_{\mathbf{E}}^{\mathbf{I}}} - \mathrm{P}_{\mathbf{E}}^{\mathbf{I}}| < \epsilon$ for some small $\epsilon > 0$, provides a relaxed consolidation constraint for noisy systems. SCM constitute a well suited framework for representing causal knowledge in the form of graphical models. The ability to trace effects through structural equations that yield explanations about the role of variables within causal models is required to make results accessible to non-experts. Actual causation, and only recently, causal abstractions and constrained causal models have come to attention in the field of causality [Halpern, 2016, Zennaro et al., 2023, Beckers and Halpern, 2019, Blom et al., 2020] and might be beneficial for future works on consolidation. Apart from computational advantages, consolidation of SCMs presents itself as a method that enables researchers to break down complex structures and present aspects of causal systems in a broadly accessible manner. Without such tools, SCMs run the danger of being only useful to specialized experts.

## Acknowledgments and Disclosure of Funding

The authors acknowledge the support of the German Science Foundation (DFG) project "Causality, Argumentation, and Machine Learning" (CAML2, KE 1686/3-2) of the SPP 1999 "Robust Argumentation Machines" (RATIO). The work was supported by the Hessian Ministry of Higher Education Research, Science and the Arts (HMWK) via the DEPTH group CAUSE of the Hessian Center for AI (hessian.ai). This work was partly funded by the ICT-48 Network of AI Research Excellence Center "TAILOR" (EU Horizon 2020, GA No 952215) and by the Federal Ministry of Education and Research (BMBF; project "PlexPlain", FKZ 01IS19081). It benefited from the Hessian research priority programme LOEWE within the project WhiteBox, the HMWK cluster project "The Third Wave of AI." and the Collaboration Lab "AI in Construction" (AICO) of the TU Darmstadt and HOCHTIEF. We thank the anonymous reviewers for their valuable feedback and constructive criticism, which significantly improved this paper. We sincerely appreciate their time and expertise in evaluating our research. We acknowledge the use of ChatGPT for restructuring some sentences while writing the paper.

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

# Supplementary Material
## "Do Not Marginalize Mechanisms, Rather Consolidate!"

## A  Evaluation of Partitioned SCM

A partitioned SCM $\mathcal{M}_{\mathcal{A}}$ consists of several sub SCM $\mathcal{M}_{\mathbf{A}}$, that, in sum, cover all variables and structural equations of an initial SCM $\mathcal{M}$. Thus, evaluation of a partitioned SCM yields the same set of values $\mathbf{v} \in \mathbf{V}$ as the original $\mathcal{M}$. Similar to the evaluation of structural equation in the initial $\mathcal{M}$, sub SCM need to be evaluated in a specific order to guarantee all $\mathbf{u} \in \mathcal{M}'_{\mathbf{U}}$ exist. As such, sub SCM can be considered multivariate variables that establish another high-level DAG. The evaluation order is determined via the relation $\mathrm{R}_{\mathbf{X}}$ as defined in Sec. 3.1 and depends on the graph partition $\mathcal{A}$ and the order of $\mathbf{X}$ imposed by the the initial SCM.

---

**Algorithm 2** Evaluation of partitioned SCM

---

 1: **procedure** PARTITIONEDSCMEVAL($\mathcal{M}_{\mathcal{A}}, \mathbf{u}, \mathbf{I}$)
 2:    $\mathbf{x} \leftarrow \mathbf{u}$                  ▷ $\mathbf{x}$ will gradually collect all values $\mathbf{x} \in \mathbf{X}$ of $\mathcal{M}$
 3:    **for** $\mathbf{A}$ **in** sort($\mathcal{A}, \mathrm{R}_{\mathbf{X}}$) **do**          ▷ Sort clusters by strict partial order imposed by $\mathcal{M}$
 4:        $\mathcal{M}'_{\mathbf{A}} \leftarrow \mathcal{M}'_{\mathbf{A}'} \in \mathcal{M}_{\mathcal{A}}$ where $\mathbf{A}' = \mathbf{A}$
 5:        $\mathbf{u}' \leftarrow \{x_i \in \mathbf{x} \mid \mathbf{X}_i \in \mathcal{M}'_{\mathbf{U}}\}$
 6:        $\mathbf{I}' \leftarrow \psi_{\mathbf{A}}(\mathbf{I})$
 7:        $\mathbf{v} = \mathcal{M}'^{\mathbf{I}'}_{\mathbf{A}}(\mathbf{u}')$
 8:        $\mathbf{x} = \mathbf{x} \cup \mathbf{v}$
 9:    **end for**
10:    $\mathbf{v} = \{x_i \in \mathbf{x} \mid \mathbf{X}_i \in \mathcal{M}'_{\mathbf{V}}\}$                  ▷ Filter all $\mathbf{u} \in \mathbf{U}$ to get $\mathbf{v} \in \mathbf{V}$
11:    **return** $\mathbf{v}$
12: **end procedure**

---

Algorithm 2 shows the evaluation of partitioned SCM, where $\mathcal{M}_{\mathcal{A}}$ is the partitioned SCM we want to evaluate, $\mathbf{u}$ are the values of exogenous variables to the initial model $\mathcal{M}$ and $\mathbf{I}$ is the set of applied interventions. The outcomes of sub SCM that are not related via $\mathrm{R}_{\mathbf{X}}$ are invariant to the evaluation order among each other. Even though $\mathrm{R}_{\mathbf{X}}$ defines the ordering of sub SCM only up to some partial order, sort($\mathcal{A}, \mathrm{R}_{\mathbf{X}}$) can pick any total ordering that is valid with $\mathrm{R}_{\mathbf{X}}$.

**Proof 1 (Consistency of Partitioned SCM Evaluation)** *Evaluations of $\mathcal{M}'_{\mathbf{A}}$ every, in step 7, compute all variables $\mathbf{V}_i \in \mathbf{A}$ by evaluating $f_i$ of the original SCM, yielding the same values as the evaluation of $\mathbf{A}$ in $\mathcal{M}$. Therefore $\mathrm{P}_{\mathcal{M}'_{\mathbf{A}}} = \mathrm{P}_{\mathcal{M}_{\mathbf{A}}}$. By Def. 4 every variable $V \in \mathbf{V}$ is contained within some sub SCM $\mathcal{M}'_{\mathbf{A}}$. The evaluation of* `PartitionedSCMEval` *is complete, in the sense that all $\mathbf{V} = \bigcup \mathcal{A} = \bigcup_{\mathbf{A} \in \mathcal{A}} \mathbf{A}$ are evaluated, as the evaluation of all $\mathcal{M}'_{\mathbf{A}} \in \mathcal{M}_{\mathcal{A}}$ is guaranteed by iterating over all $\mathbf{A}$ in step 2. Finally $\mathrm{P}_{\mathcal{M}'_{\mathcal{A}}} = \bigcup_{\mathbf{A} \in \mathcal{A}} \mathrm{P}_{\mathcal{M}'_{\mathbf{A}}} = \bigcup_{\mathbf{A} \in \mathcal{A}} \mathrm{P}_{\mathcal{M}_{\mathbf{A}}} = \mathrm{P}_{\mathcal{M}_{\mathbf{V}}}$.* ∎

## B  Complexity reduction in function composition

Reduction of encoding length might vary depending on the type and structure of the equations under consideration. No compression of structural equation is gained when the system of consolidated equations is already minimal. Compression of equation to an identity function is showcased in the following.

### B.1  Compression of chained inverses

Reduction to constant complexity for the unintervened system is reached in the case of $f_B = f_A^{-1}$. Consider the equation chain of $X \to A \to B$ with $A$ getting marginalized. Immediately $f'_B :=  f_B \circ f_A = f_A^{-1} \circ f_A = \mathrm{Id}$ follows. Therefore, $B := X$, which is a single assignment of the value(s) of $X$ into $B$. Remaining complexity within the consolidated function is then only due to conditional branching in cases of $do(A = a), do(B = b) \in \mathbf{I}$.

## B.2  Matrix composition is not sufficient for compressing equations

The operation of matrix multiplication, as a way of expressing composition of linear functions, stays within the class of matrices. Matrix multiplication, therefore, serves as a possible candidate to be considered when consolidating equations and reducing the encoding length of a linear structural systems. When written down an a 'high-level' view, matrices can expressed in terms of single variables $A, B \in \mathbb{R}^{M \times N}$ and matrix multiplication $\times : \mathbb{R}^{M \times N} \times \mathbb{R}^{N \times O} \to \mathbb{R}^{M \times O}$. Assuming equations $f_Y := A \times X$ and $f_Z := B \times X$, we can reduce the length of the composed equation $f'_Z := A \times B \times X$ by multiply the matrices $A$ and $B$ together, $f_i = C \times X$ with $C = A \times B$. While we effectively reduced the number of high-level symbols written in the equation, we are hiding computational complexity in the structure of the matrix $C$. The following simple counterexample demonstrates a situation where the size, as well as, the number of non-zero entries even increases:

$$
\begin{array}{ccc}
C & A & B \\
\begin{bmatrix} 0 & 1 & 1 \\ 0 & 1 & 1 \\ 0 & 1 & 1 \end{bmatrix} = & \begin{bmatrix} 0 & 1 \\ 0 & 1 \\ 0 & 1 \end{bmatrix} \times & \begin{bmatrix} 0 & 0 & 0 \\ 0 & 1 & 1 \end{bmatrix}
\end{array}
$$

Thus, proving that pure matrix multiplication, is not suitable to keep, or even minimize, the size of composed function representations.

## B.3  Compression over Finite Discrete Domains

Consolidation may reduce the number of variables within a graph, but burdens the remaining equations with the complexity of the consolidated variables. Without the need to explicitly compute values of consolidated variables, we might leverage cancellation effects to simplify equations, as outlined in the main paper. In terms of compression, no guarantees can be given in the general case. However, we will now show, that the often considered case of chained maps between finite discrete domains simplifies or at least preserves complexity.

The cardinality of the image of a deterministic function $f : \mathcal{X} \to \mathcal{Y}$ between two finite discrete sets $\mathcal{X}, \mathcal{Y}$ is bounded by the cardinality of its domain: $|\operatorname{Img}(f)| \leq |\operatorname{Dom}(f)| \leq |\mathcal{X}|$, where $\operatorname{Img}(f)$ is the image and $\operatorname{Dom}(f)$ the domain of $f$. In particular, the strict inequality $|\operatorname{Img}(f)| < |\operatorname{Dom}(f)|$ holds for all non-injective maps. Function composition may further reduce the 'effective' domain $\operatorname{Dom}_{\text{effective}}(f)$ of a function, by only considering values of the image of the previous map as inputs to the next function. In contrast considering to all possible values of $\mathcal{X}$ in the case of the non-composed map, the image of the previous function may only be a subset of $\mathcal{X}$. Therefore, $f_2 \circ f_1 \Rightarrow |\operatorname{Img}_{\text{effective}}(f_2)| \leq |\operatorname{Dom}_{\text{effective}}(f_2)| = |\operatorname{Img}(f_1)| \leq |\operatorname{Dom}(f_1)|$. In particular, the effective image of a composition chain $f_n \circ \cdots \circ f_1$ is bounded by the function with the smallest image: $|\operatorname{Img}_{\text{effective}}(f_n \circ \cdots \circ f_1)| \leq \min |\operatorname{Img}(f_i)|$. Thus, equation chains over finite discrete domains strictly preserve or reduce the effective size of the image, allowing for a possibly simpler combined representation in comparison to representing the functions individually.

## C  Reparameterization of non-deterministic structural equations.

Consolidation of structural equations might lead to duplication of non-deterministic terms within consolidated systems. For example when consolidating fork structures (compare to Sec. 4.1). Without further precautions, different values might be sampled from the duplicated non-deterministic equations. An example where consolidating a variable $B$ with a non-deterministic equation $f_B$ (indicated by a squiggly line) leads to inconsistent behaviour is shown in 6. In $\mathcal{M}_1$, $C$ and $D$ both copy on the value of $B$. Therefore, $c = d$ yields always. $\mathcal{M}_{1'}$ shows a graph where $B$ is consolidated from $\mathcal{M}_1$. As a result the non-deterministic equation $f_B$ is duplicated into the equations of $C$ and $D$, such that $f_C := \operatorname{Bern}(A)$ and $f_D := \operatorname{Bern}(A)$. Within the consolidated model $\mathcal{M}_{1'}$ different values might be be sampled from the different noise terms $\operatorname{Bern}(A)$ in $f_C$ and $f_D$. Consequently $c \neq d$ might occur in $\mathcal{M}_{1'}$. To obtain consistent behaviour with the initial $\mathcal{M}_1$, we need to ensure agreement about the value of $\operatorname{Bern}(A)$ across all instances of the duplicated equation. To do so, we reparameterize $\mathcal{M}_1$ and explicitly store a fixed value, sampled from $\operatorname{Bern}(A)$, into a new exogenous variable $R$. The equation $f_B$ is then reparameterized into a deterministic structural equation taking the variable $R$ as an additional argument, resulting in $\mathcal{M}_2$. When consolidating $B$ within $\mathcal{M}_2$, all instances of $f_B$ now yield the same value, as the noise term is fixed via $R$ and finally $P_{\mathcal{M}'_2} = P_{\mathcal{M}_1}$.

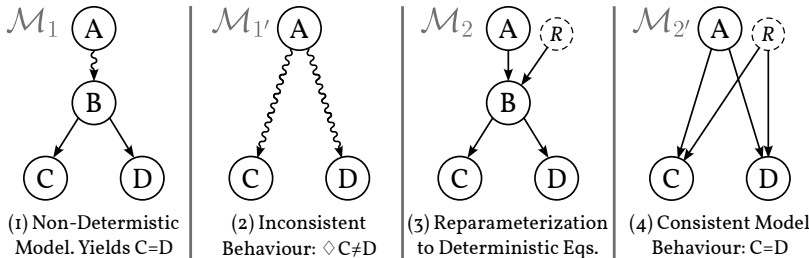

| (1) Non-Determistic Model. Yields C=D | (2) Inconsistent Behaviour: ◇C≠D | (3) Reparameterization to Deterministic Eqs. | (4) Consistent Model Behaviour: C=D |

Figure 6: **Reparameterization of non-deterministic models.** The SCM $\mathcal{M}_1$ contains a non-deterministic equation $B := \mathrm{Bern}(A)$ (marked with a squiggly line). With $C := B$ and $D := B$, $\mathcal{M}_1$ always yields $C = D$. Simply consolidating (or marginalizing) $B$ creates a model $\mathcal{M}_{1'}$ with $C := \mathrm{Bern}(A)$ and $D := \mathrm{Bern}(A)$, such that possibly $C \neq D$. Reparameterizing $f_B$ by introducing an exogenous random variable $R := \mathcal{U}(0,1)$ and $B := A < R$, yields the SCM $\mathcal{M}_2$ with only deterministic equations. Consolidating (or marginalizing) $B$ in $\mathcal{M}_2$ leads to $\mathcal{M}_{2'}$ where $C := A < R$ and $D := A < R$, thus always $C = D$.

## D  Consolidation Examples

In this section we show further detailed applications of consolidation. Section D.1 presents the worked out consolidation of the dominoes motivating example of the paper, with regard to generalizing abilities of consolidates models. Section D.2 considers consolidation of the classical firing squad example. In contrast to the other examples, we focus on consolidating graphs with multiple edges in the causal graph. Lastly we provide the causal graph and structural equations of the game agent policy discussed in the main paper, in Section D.4.

### D.1  Motivating Example: Dominoes

While we applied consolidation to a particular SCMs in the main paper, we will discuss the motivating example with focus on obtaining representations that cover generalize over populations of SCM. We demonstrate this on the particular example of a rows of dominoes, as a simple SCM with highly homogenous structure. Regardless of whether the SCM is obtained by using methods for direct identification of causal graphs from image data, as presented by Brehmer et al. [2022], or abstracting physical simulation using $\tau$-abstractions [Beckers and Halpern, 2019]; we assume to be provided with a binary representation of the domino stones. The state of every domino $S_i$ indicates whether it is standing up or getting pushed over. In this case, the structural equations for all dominoes are the same: $f_i := S_{i-1}$. As a result tipping over the first stone in a row will lead to all stones falling. Also, we are only interested in the final outcome of the chain. That is, whether the last stone will fall or not ($\mathbf{E} = \{S_n\}$). Again, we use consolidation to collapse the structural equations in the unintervened case: $S_n := f_n \circ \cdots \circ f_1 := S_1$. We consider a single active allowed intervention of holding up any of the dominoes or tipping it over, $\mathcal{I} = \{do(S_i = 0), do(S_i = 1)\}$. Upon evaluation, the unconsolidated model needs to check for every domino if it is being intervened or not, requiring $n$ conditional branches. Using the fact that perfect interventions 'overwrite' the variable state for the following dominoes, we introduce a first order quantifier that handles all intervention in a unified way. Finally, by combining the formulas of the intervened and unintervened case, we find the following simple equation:

$$S_n := \begin{cases} x_i & \text{if } \exists\, do(S_i = x_i) \in \mathbf{I} \\ S_1 & \text{else} \end{cases}$$

The resulting equation no longer has a notion of the actual number of dominoes and, in fact, it is invariant to it. We realise that introducing the first-order for-all $\forall$ and exists $\exists$ quantifiers allows for a unified representation of arbitrary chains of dominoes. Similar observations are discussed in Peters and Halpern [2021] and Halpern and Peters [2022] which introduce generalized SEM (GSEM). As intermediate the equations are no longer computed explicitly, the structural equations of consolidated models for different row lengths only differ in the set of allowed interventions $\mathcal{I}$. That is, for a row of three domino stones $\mathcal{I} = \{do(V_1 = v_1), do(V_2 = v_1), do(V_3 = v_1)\}$, while for four stones the additional $do(V_4 = v_1)$ is defined. As set out in the introduction of this paper, we consider

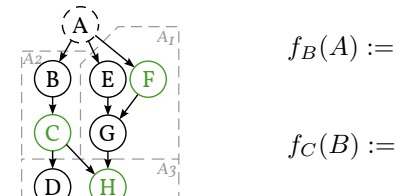

$$f_B(A) := \begin{cases} 0 & \text{if } A \le 5 \\ 1 & \text{if } A > 5 \end{cases}$$

$$f_C(B) := \begin{cases} \text{true} & \text{if } B = 0 \\ \text{false} & \text{if } 0 \le B \le 10 \\ \text{true} & \text{otherwise} \end{cases}$$

$$f_E(A) := A\%5 = 0$$
$$f_F(A) := A\%10 = 0$$
$$f_G(E, F) := E \land F$$
$$f_D(C) := \neg C$$
$$f_H(C, G) := C \lor G$$

Figure 7: **Example SCM for the Application of CONSOLIDATE.** The figure shows a toy SCM for demonstrating application of the CONSOLIDATE algorithm. Consider the following SCM with its structural equations and resulting graph (endogenous variables are $B, C, D, E, F, G, H$ with only one exogenous $A$ with each structural equation highlighted on the r.h.s., note that the subscript on $f\_$ denotes the variable to be determined e.g. $B \leftarrow f_B(A)$). In the first step, the algorithm's user decides on a partition. Let's consider for instance the following partition i.e., allowed intervention and consolidation sets: $\mathcal{A} = \{\{E, F, G\}, \{B, C\}, \{D, H\}\}; \mathbf{E} = \{C, F, H\}; \mathcal{I} = \{\{do(D = \text{true})\}, \{do(D = \text{false})\}, \{do(G = \text{false})\}\}$.

consolidation as a tool for obtaining more interpretable SCM. Towards this end, consolidation might help us in detecting similar structures within an SCM. Doing so eases understanding of causal systems, as the user only has to understand the general mechanisms of a particular SCM once and is then able to apply the gained knowledge to all newly appearing SCM of the same type.

### D.2 Firing Squad Example

While the dominoes and tool wear examples where mainly considering the consolidation of sequential structures, we want to briefly demonstrate the consolidation of structural equations that are arranged in a parallel fashion. We consider a variation of the well known firing squad example [Hopkins and Pearl, 2007] with a variable number $N$ of rifleman. A commander ($C$) gives orders to rifleman ($R_i, i \in \{1 \ldots N\}$), which shoot accurately and the prisoner ($P$) dies. For the sequential stacking of equations we found that interventions exert an 'overwriting' effect. That is, every intervention fixes the value of a variable, making the unfolding of the following equations independent of all previous computations. To yield a similar effect for parallel equations we need to block *all* paths between the cause and effect. In this scenario, this can easily be expressed by using an all-quantifier. When consolidating the SCM, we consider only the captain $C$ and prisoner $P$, $\mathbf{E} = \{C, P\}$, while allowing for any combination of interventions that prevent the rifleman from shooting $\mathcal{I} = \mathcal{P}(\{do(R_i = 0)\}_{i \in \{1 \ldots N\}})$. After consolidation, we obtain the following equation:

$$P := \begin{cases} \text{lives} & \text{if } C = 0 \lor (\forall S_i. \, do(S_i = 0) \in \mathbf{I}) \\ \text{dies} & \text{else} \end{cases}$$

As with the dominoes example, we are again in a situation where the consolidated equation intuitively summarizes the effects of individual: "The prisoner lives if the captain does not give orders, or if all riflemen are prevented from shooting".

### D.3 Step-by-step CONSOLIDATE Application

In this section we provide a step-by-step application of the CONSOLIDATE algorithm given in Algorithm 1. Consider the SCM shown in Figure 7 with its structural equations and resulting graph. The endogenous variables are $B, C, D, E, F, G, H$ with only one exogenous $A$. Structural equation are highlighted on the right-hand side. Note that the subscript on $f_x$ denotes the variable to be determined e.g. $B \leftarrow f_B(A)$.

In a first step, the algorithm's user has to decide on a suitable partition. Consider for instance the following partition (indicated by dashed lines in the figure), the following allowed intervention and consolidation set:

$\mathcal{A} = \{\{E, F, G\}, \{B, C\}, \{D, H\}\}$
$\mathcal{I} = \{\{do(D = \text{true})\}, \{do(D = \text{false})\}, \{do(G = \text{false})\}\}$
$\mathbf{E} = \{C, F, H\}$

The following example presents a step-by-step application of the CONSOLIDATE algorithm for the cluster $\mathbf{A}_1 = \{E, F, G\}$:

Step 3: $\qquad \mathbf{E}_1 \leftarrow \{E, F, G\} \cap \{C, F, H\} = \{F\}$

Step 4: $\qquad \mathbf{E}_1' \leftarrow \{F\} \cup (\text{pa}(\mathbf{V} \setminus \{E, F, G\}) \cap \{E, F, G\})$
$\qquad\qquad\qquad = \{F\} \cup (\{A, B, C, G\} \cap \{E, F, G\}) = \{F, G\}$

Step 5: $\qquad \mathbf{U}_{\mathbf{A}_1} \leftarrow \text{pa}(\{E, F, G\}) \setminus \{E, F, G\} = \{A, E, F\} \setminus \{E, F, G\} = \{A\}$

Step 6: $\qquad \mathcal{I}_{\mathbf{A}_1} \leftarrow \{\{do(X_i = v) \in \mathbf{I} : X_i \in \{E, F, G\}\} : \mathbf{I} \in \mathcal{I}\} = \{\{do(G = \text{false})\}\}$

Step 7: $\qquad \rho_{\mathbf{E}_1'} \leftarrow \{f_E(A) := A \bmod 5 = 0;$
$\qquad\qquad\qquad f_F(A) := A \bmod 10 = 0;$
$\qquad\qquad\qquad f_G(E, F) := E \wedge F\}$

Step 8: $\qquad \rho_{\mathbf{E}_1'}^{\star} \leftarrow \text{argmin}\mathcal{K}(\rho_{\mathbf{E}_1'}) = \{$
$\qquad\qquad\qquad \rho_F(A) := A \bmod 10 = 0;$
$\qquad\qquad\qquad \rho_G(F, \mathbf{I}_{\mathbf{A}_1}) := F \wedge (do(G = \text{false}) \notin \mathbf{I}_{\mathbf{A}_1})\}$

Step 9: $\qquad \mathcal{M}_{\mathbf{A}_1, \mathbf{E}} \leftarrow (\{F, G\}, \{F\}, \rho_{\mathbf{E}_1'}^{\star}, \{\{do(G = \text{false})\}\}, P_A)$

Note how computing $f_E$ is no longer required. In a similar fashion, equations in $\mathbf{A}_2$ resemble a chain that can be composed: $f_C \circ f_B$ (previously called '*stacked*'; cf. Sec. 4.1). Since $|\text{Img}(f_B)| = 2$, at least one of the three conditions of $f_C$ (since $f_C$ is a 3-case function) will be discarded. (Eventually yielding $\rho_{\mathbf{E}_2'}^{\star} \leftarrow \{\rho_C(A) := A \leq 5\}$). As $D$ is not in $\mathbf{E}$ and not required by any other sub SCM it can be marginalized. $A_3$ then reduces to $\rho_{\mathbf{E}_3'}^{\star} \leftarrow \{\rho_H(C, G) := C \vee G\}$.

### D.4 Revealing Agent Policy: Causal Graph and Equations

In this section we explicitly list the structural equations representing observed interactions between a platformer environment and a possible rule based agent. The resulting causal graph is shown in Fig.8 at the end of the appendix. Except for the parentless variables 'coin_reward', 'powerup_reward', 'enemy_reward', 'flag_reward', 'player_position', 'position_coin', 'position_powerup', 'position_enemy', 'position_flag' and 'target_flag', which are exogenous and determined by the environment, all variables are considered endogenous:

$\text{player\_position}, \text{position\_coin}, \text{position\_powerup}, \text{position\_enemy}, \text{position\_flag} \in [0..1]^2$

$\text{coin\_reward} := 3; \text{powerup\_reward} := 1; \text{enemy\_reward} := 9; \text{flag\_reward} := 2$

With $X$ in $\{\text{coin}, \text{powerup}, \text{enemy}, \text{flag}\}$ :

$\qquad \text{distance\_}X := \|\text{position\_}X - \text{player\_position\_}X\|_2$

$\qquad \text{near\_}X := \text{distance\_}X < 3.0$

$\qquad \text{targeting\_cost\_}X := 1.0 + 0.5 \times \text{distance\_}X$

$\text{target\_coin} := \text{targeting\_cost\_coin} < \text{enemy\_reward}$

$\text{target\_powerup} := \text{targeting\_cost\_powerup} < \text{powerup\_reward}$

$\text{target\_enemy} := \text{targeting\_cost\_enemy} < \text{enemy\_reward} \wedge \text{powered\_up}$

$\text{target\_flag} := \text{True}$

$\text{powered\_up} := \text{target\_powerup}$

$\text{towards\_coin} := \text{target\_coin} \wedge \text{coin\_reward} > \max(\{X\_\text{reward}|\text{target\_}X\}_{X \in \{\text{powerup}, \text{enemy}, \text{flag}\}})$

$\text{towards\_powerup} := \text{target\_powerup} \wedge \text{powerup\_reward} > \max(\{X\_\text{reward}|\text{target\_}X\}_{X \in \{\text{coin}, \text{enemy}, \text{flag}\}})$

$\text{towards\_enemy} := \text{target\_enemy} \wedge \text{enemy\_reward} > \max(\{X\_\text{reward}|\text{target\_}X\}_{X \in \{\text{enemy}, \text{powerup}, \text{flag}\}})$

$\text{towards\_flag} := \text{target\_flag} \wedge \text{flag\_reward} > \max(\{X\_\text{reward}|\text{target\_}X\}_{X \in \{\text{coin}, \text{powerup}, \text{enemy}\}})$

$\text{jump} := \text{near\_enemy} \wedge \neg\text{powered\_up}$

$$
\text{planning\_sequence}_i := 
\begin{cases}
\text{finished} & \text{if towards\_flag} & \wedge\ (\text{flag} & \in \bigcup_{j=1}^{i-1} \text{planning\_sequence}\_j) \\[1.5em]
\text{coin} & \text{if towards\_coin} & \wedge\ (\text{coin} & \notin \bigcup_{j=1}^{i-1} \text{planning\_sequence}\_j) \\[1.5em]
\text{powerup} & \text{if towards\_powerup} & \wedge\ (\text{powerup} & \notin \bigcup_{j=1}^{i-1} \text{planning\_sequence}\_j) \\[1.5em]
\text{enemy} & \text{if towards\_enemy} & \wedge\ (\text{enemy} & \notin \bigcup_{j=1}^{i-1} \text{planning\_sequence}\_j) \\[1.5em]
\text{flag} & \text{if towards\_flag} & \wedge\ (\text{flag} & \notin \bigcup_{j=1}^{i-1} \text{planning\_sequence}\_j) \\[1.5em]
\text{finished} & \text{else} &&
\end{cases}
$$

$$
\begin{aligned}
\text{score} := 20 - \text{time\_taken} \\
+\ \text{coin\_reward if coin} \in \text{planning\_sequence}_i \\
+\ \text{powerup\_reward if powerup} \in \text{planning\_sequence}_i \\
+\ \text{enemy\_reward if enemy} \in \text{planning\_sequence}_i \wedge \text{powerup} \in \text{planning\_sequence}_i \\
+\ \text{flag\_reward if flag} \in \text{planning\_sequence}_i
\end{aligned}
$$

# E   Mathematical symbols and notation

The following table contains mathematical functions and notation used throughout the paper.

| Notation | Meaning |
|---|---|
| $X;\ \mathbf{X}$ | A (set of) variable(s). |
| $x;\ \mathbf{x}$ | Value(s) of $X;\ \mathbf{X}$. |
| $\mathbf{X}_i$ | The i-th variable of $\mathbf{X}$. |
| $\mathbf{X_S}$ | The subset $\{\mathbf{X}_i : i \in \mathbf{S}\}$ of $\mathbf{X}$. |
| $\mathrm{P}_{\mathbf{X}}$ | A probability distribution over variables $\mathbf{X}$. |
| $x \sim \mathrm{P}_X$ | A value $x$ sampled from a distribution over $X$. |
| $\mathcal{P}(\cdot)$ | The power set. |
| $f \circ g$ | Function composition, $(f \circ g)(x) = f(g(x))$. |
| $\prod_{X_i \in \mathbf{X}} \mathcal{X}_i$ | N-ary Cartesian product over the domain of $\mathbf{X}$. |
| $\lVert \cdot \rVert_2$ | $l^2$ vector norm. |
| $\mathcal{U}(a, b)$ | Uniform Distribution. |
| $\mathcal{N}(\mu, \sigma^2)$ | Normal Distribution. |
| $\mathrm{Bern}(p)$ | Bernoulli distribution; Takes value 1 with probability $p$ and 0 otherwise. |
| $\mathrm{P}_{\mathcal{M}}$ | Probability distribution over the SCM $\mathcal{M}$. |
| $\mathrm{P}_{\mathcal{M}}^{\mathbf{I}}$ | Probability distribution over the SCM $\mathcal{M}$ under intervention $\mathbf{I}$. |
| $V_i$ | An endogenous variable of an SCM $\mathcal{M}$. |
| $U_i$ | An exogenous variable of an SCM $\mathcal{M}$. |
| $f_i$ | Structural equation of the variable $X_i$. |

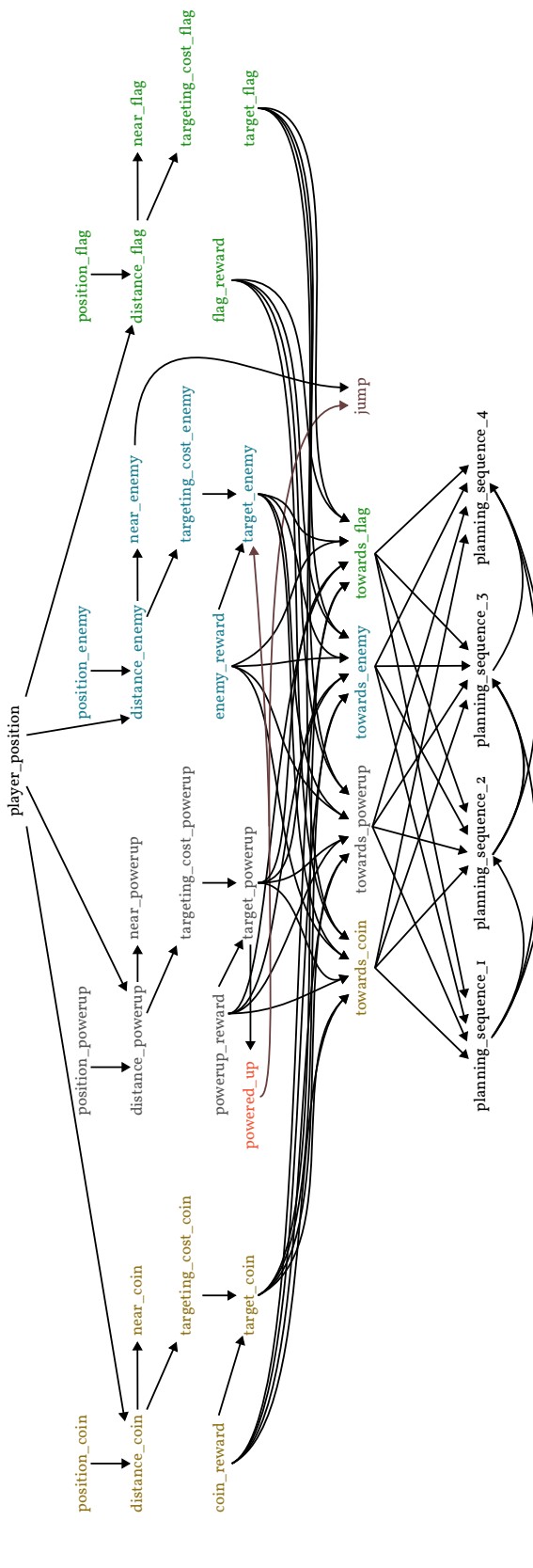

Figure 8: **Causal graph of an agent policy.** The causal graph of a greedy agent inside an platformer environment. The parentless variables are exogenous. Their value is determined via the game environment. The final 'score' variable is left out for clarity.

