# OpenReview forum: "Do Not Marginalize Mechanisms, Rather Consolidate!"
_NeurIPS.cc/2023/Conference — NeurIPS 2023 poster_

### Official Review · Reviewer_ZzBw · 2023-07-04

**Soundness:** 2 fair
**Presentation:** 3 good
**Contribution:** 3 good
**Rating:** 7
**Confidence:** 3

**Summary:**

The paper presents a framework for causal reasoning that supports the simplification of large structural causal models (SCMs).  One key operation that supports this simplification is consolidation of a SCM such that only a subset of endogenous variables are explicitly modelled, but in such a way that all possible interventions are still supported. Another key operation is the partitioning of an SCM into sub-models, in such a way that endogenous variables from one sub-SCM act as exogenous variables in another.

Potential reductions in complexity are then discussed by the use of these operations and associated constraints. For instance, replacing equations within SCMs with computationally simpler expressions,  and dropping equations which do not relate to variables of interest. Throughout there is a focus on retaining information about the effects of interventions. The paper concludes with two examples: one on modelling tool wear on a milling machine and another, more complex example, exploring planning policies for a simple platformer game.


**Strengths:**

* The paper very clearly presents its contribution and relates this well to existing work, justifying the relevance of the contribution to the field.
* The paper builds its arguments formally, and with intuition, and presents meaningful relevant results and constraints.
* The examples illustrate well the benefits of the proposed framework.
* The contributions appear meaningful to me and likely to be of relevance to other researchers, particularly those concerned with large scale causal modelling.



**Weaknesses:**

* At times the arguments are a little vague or the explanations incomplete.
* Although mostly clear, the formal notation sometimes leaves a little to be desired.
* There are a few places where the authors appear to make errors or omissions in their explanations.

Issues with understanding:
1. In definition 1, the set of possible interventions is a little unclear. There appears to be 1 possible intervention, $I_i$ per endogenous variable, $X_i$. Or can there be multiple potential, but mutually exclusive, interventions per endogenous variable? What exactly is $I_i$?
2. On line 94 $f_i(\textbf{x}, x_0)$ is used to indicate the structural equation for variable $X_i$ under the intervention that $X_j$ is set to value $x_0$, but this seems a little under-defined to me.
3. On line 98, the authors state that $\mathcal{M}$ entails infinitely many intervened distributions, but this seems to conflict with earlier notation (see 1.).
4. Definition 2 could do with an explanation of $P^{\textbf{I}}_{\textbf{E}}$. I am assuming this means the distribution of target variables in $\textbf{E}$ under some intervention $\textbf{I}$ but I can't see this stated anywhere.
5. The notation $\rho(\textbf{U},\textbf{I})$ first introduced in Definition 2 does not refer to the subset of variables $\textbf{E}$ to which it refers. This would be good practice anyway, but becomes more problematic when partially consolidated SCMs are considered (Def 5) as the variables of interest $\textbf{E}$ are augmented with additional variables that act as exogenous variables in other sub-SCMs.
6. The caption for Figure 2 (right) states that the dotted line indicates explicit computation for $X_2$ but it isn't clear what is meant by this.
7. In section 3, in a number of places (lines 159, 165, 187), there appears to be repeated errors in the notation, e.g. $\textbf{V} \in \textbf{A}$. I think that $\textbf{V}$ is the complete set of endogenous elements while $\textbf{A}$ is a subset of $\textbf{V}$
8. In lines 175-183, the partitioning of an SCM appears to  require that exogenous variables $\textbf{U}_i$ of a sub SCM must be endogenous variables of another sub-SCM in the same partitioned SCM. But could they be truly exogenous variables of the whole system? Also, there appears to be notational irregularities in this paragraph relating to what is a sub-SCM and what is a partitioned SCM.
9. Things get a bit messy around definition 5 with respect to $\textbf{E}$ and $\textbf{E}'$. The distinction between these two sets of variables could be clearer. For instance, if I am considering partially consolideted SCM $\mathcal{M}_{\mathcal{A},\textbf{E}'}$ then how do I know what the set $\textbf{E}$ is that is used to define $\textbf{E}'$.
10. In section 4, there is a chain of inference on lines 234-235 that is difficult to follow (what is the scope of the universal and existential quantifiers?) and some entity $D$ appears without being defined.
11. I got a little lost in section 4.1. In particular, the discussion of conditional branching and stacking was a little vague.

## Post rebuttal

After reading the rebuttal and individual responses to the above points, I am raising my recommendation to "accept".

**Questions:**

My questions have mostly been articulated in the **Weaknesses** field.


**Limitations:**

The authors discuss limitations and the possible relaxations of these effectively.

---

> ### Author Rebuttal · Authors · 2023-08-09
>
> Thank you reviewer ZzBw for the detailed comments in your review which have led to improving our paper to its new version. In the following we go over each of the highlighted points one-by-one.
>
> * *Regarding the set of possible interventions:*
>
>   Thank you for pointing this out, as it helped improve as to avoid future confusion for other readers. In our paper we refer to an *intervention* $I$ as a specific instantiation of the do operator on a variable, e.g. $do(X_0 = 5)$. While sets of multiple interventions $\textbf{I}$ (note the different formatting) can be applied over the whole SCM. In order to restrict notational clutter, we assumed that the set of allowed interventions is countable. Such that we get something like $I_0 = do(X_0=0); \dots; I_5 = do(X_0=5); I_6 = do(X_1=0); \dots$ for instance. While reviewing your comment we found that our formalism might break down for SCM containing uncountable (e.g. real-valued) variables (and did not restrict the actual sets to only contain one intervention per variable). To fix our mistake (and keep notational brevity), we have switched to writing $I_{i,v}$, indicating the intervention as $do(X_i = v)$. In consequence Def 1. bullet point 4 is rewritten as:
>
>   "$\mathcal{I} \subseteq \bigcup_{i \in [1\dots N]} \bigcup_{v \in {\mathcal{X}\_i}}$ $\{I_{i,v}\}$ such that $\forall \mathbf{I} \in \mathcal{I}. \forall i \in [1\dots N].(\exists! v\in\mathcal{X}\_i. \mathbf{I}\_{i,v} \in \mathcal{I}) \lor (\lnot \exists v \in \mathcal{X}\_i. \mathbf{I}\_{i,v} \in \mathcal{I})$ and $\mathbf{J} \subset \mathbf{I} \in \mathcal{I} \rightarrow \mathbf{J} \in \mathcal{I}$. $\mathbf{I}$ is the set of perfect interventions under consideration. A perfect intervention $\text{do}(V_i = v_i)$ replaces the unintervened $f_i$ by the constant assignment $V_i := v_i$.
>
> * *Regarding the "explicit computation" in Fig.2:*
>
>   With the figure we want to express that the value of $X_2$, as an element in $\textbf{E}'$, is computed by $\rho_{\mathbf{E}'}$ (and there would be no way to depict the variable in the figure otherwise). In contrast to `normal' variables one can not directly intervene on this variable by simply cutting the edge to $\rho_{\mathbf{E}'}$ via an intervention. As stated in the caption, $X_2$ is the output of $\rho_{\mathbf{E}'}$ and not a normal variable. With 'explicit' we indicate that its value is still visible to the user. In contrast, $\rho_{\mathbf{E}'}$ might also compute the value of $X_1$ internally. However, this depends on the compression inside $\rho_{\mathbf{E}'}$ and $X_1$ is not visible to the user. Thanks for the pointer, we've added a brief statement on this to the new paper version.
>
> * *Regarding (7). Notational error on $\mathbf{V},\mathbf{A}$*:
>
>   Thanks for spotting that! Correct, we've fixed this.
>
> * *Regarding (8): Truly exogenous variables*
>
>   Indeed, they could also be truly exogenous variables. We've added a comment on this.
>
> * Regarding (8). Sub-SCM and partitioned SCM:
>
>   As defined in Definition 4 a partitioned SCM $\mathcal{M}\_{\mathcal{A}}$ consist of multiple sub SCM $\mathcal{M}\_{\mathbf{A}_i}$. For better readability we discarded the index $i$ wherever it was undefined and wrote $\mathcal{M}\_{\mathbf{A}}$ instead to mean any (unspecific) $\mathbf{A} \in \mathcal{A}$. We've added a clarifying statement.
>
> * *Regarding (9). Clarification on $\mathbf{E},\mathbf{E}'$*:
>
>   To consider either $\mathbf{E}$ or $\mathbf{E}'$ actually depends on the standpoint of the user (of the consolidation operation). From a computational perspective, $\mathbf{E}'$ is important as it contains all variables that need to be computed by $\rho$. While $\mathbf{E}$ rather captures important aspects of the SCM to the user i.e., variables of interest. However, inferring for instance that $\mathbf{E}'$ can be deduced from $\mathbf{E}$, but not the other way around would pose a strong argument. We therefore choose to refer to sub SCM with $\mathcal{M}\_{\mathbf{E}'}$ (and in the same breath write $\rho_{\mathbf{E}'}$) but use $\mathcal{M}\_{\mathcal{A},\mathbf{E}}$ to retain the initial set $\mathbf{E}$ in the overall notation. Thanks for pointing out this nuanced discussion, we've added a short paragraph.
>
> * *Regarding (11): General considerations on compressibility*
>
>   Computing minimal representation is generally not possible (as outlined in 'General compression of equation systems' of Sec. 4). Nonetheless we intended on giving a more involved perspective on some of the basic structures (chains, forks and colliders) that appear in the SCM's implied graph structure to allow for some local optimizations using our approach (even if they do not allow to arrive at the true minimal representation $f^\star_i$). Ultimately, there is no way of measuring compressibility of equations by only considering their connecting graph structure. However, even considering local substructures of graphs might improve compression. We have added this discussion to section 4.1. Thanks for raising this. In conjunction with other reviewers' comments, we furthermore present a concrete example of consolidating a collider and chain which will be added to the final paper. (Consider consolidation of $\mathbf{A}_1$ and $\mathbf{A}_2$ respectively in the provided PDF. Actual steps are provided in the answer to reviewer tWPL)
>
> **Disclaimer:** our actual answer covered all and thus more of your points. This is due to the 6000 character limit by OpenReview for NeurIPS this year. Therefore, we chose to delete points we deemed that you would maybe find less important than what we kept (We incorporated all of your comments into the paper). We can maybe provide them during the author-reviewer discussions.
>
> We would like to once again sincerely thank you for thoroughly checking and commenting on our work!
> It has greatly improved the overall soundness and quality of our paper in its new version and we look forward to some possible further discussions with you.
> Kind regards, your authors.

---

> > ### Comment · Reviewer_ZzBw · 2023-08-14
> >
> > Thank you for answering my questions. I think I can understand most of these now. I wanted to make the following points:
> >
> > *    Regarding the set of possible interventions:
> >
> > I think I see. If I do then there can be 0 or 1 unique intervention values v for each variable $X_i$ in some intervention set $I$, but the set of possible interventions $\curly{I}$ includes all possible values for each variable. Is that right? The set notation you use elsewhere is maybe more compact/readable than the equation in your response.
> >
> > * Regarding the "explicit computation" in Fig.2:
> >
> > I think I understand the explanation. Perhaps you could name the variable differently, e.g. $X_2'$ as it isn't the "same" variable, in the sense that it doesn't have the same properties as $X_2$. For instance, it may be the case that there is some original intervention $do(X_2=v)$, but this may not necessarily change the value of $X_2'$. Or am I misunderstanding?
> >
> > After reading the other reviews, the rebuttal and your responses to each of the reviewers I am raising my assessment to Accept.

---

> > > ### Author Response · Authors · 2023-08-15
> > >
> > > Thank you for further engaging in the discussion, here are our comments to these two points:
> > >
> > > 1. Correct. The intervention set is a subset of the set of all possible interventions, where the set of all possible interventions is defined as the union over all intervenable variables and a union over each combination of values (from the respective domains of each of the variables) that these variables can take. We have switched to the following set notation: $\mathcal{I} \subseteq \\{\\{I_{i,v\_i}\\}\_{i \subseteq \\{1\dots N\\}}\\}\_{\mathbf{v} \in {\pmb{\mathcal{X}}}}$ where $v_i$ is the i-th element of $\mathbf{v}$ and such that $\mathbf{J} \subset \mathbf{I} \in \mathcal{I} \rightarrow \mathbf{J} \in \mathcal{I}$.
> > > (Reminder: $\pmb{\mathcal{X}}$ is defined as the cartesian product covering all variable domains $\mathcal{X}_i$.) Despite the rather minor technical drawback of generating some $\mathbf{I}$ 'multiple times' we appreciate the suggestion of a more compact formalization with reduced constrains.
> > > 2. We are sorry about the confusion regarding our answers. The consolidated $X_2$ will behave exactly the same (with and without interventions). This becomes more clear when looking only at the structural equations: $X_2$ is computed via $\rho_{\mathbf{E}'}(\mathbf{U}, \mathbf{I})$, which is required to be consistent with the original $\mathcal{M}$ (- compare to Def. 2). However, when considering Figure 2 without any additional precautions, a potential reader may get the impression that there are now two possible places to intervene: (1) via parameter $\mathbf{I}$, e.g. $\mathbf{I} = \\{ do(X_2 := v)\\}$, of $\rho_{\mathbf{E}'}(\mathbf{U}, \mathbf{I})$ as it computes the value of $X_2$  and (2) via a 'classical' intervention on the graph, $do(X_2 := v')$, cutting the edge between $X_2$ and $\rho_{\mathbf{E}'}$. This can not be, as two, possibly different, intervention values for $X_2$ would lead to an inconsistent behaviour. (Note that in the second case $\rho$ would not only compute an incorrect value for $X_2$, but also incorrectly compute values for the dependent $X_4$ and $X_5$). For this reason - and while observing the value of $X_2$ - we are not allowed to manipulate the connection between $\rho_{\mathbf{E}'}$ and $X_2$ (= we do not allow intervention of case (2)). To indicate this restriction in the graph, we decided for the altered visual representation. Thanks again for asking this. We will make sure to add the main points of this discussion in the figure caption.
> > >
> > > Thank you for helping in improving our paper by providing all of this great feedback!
> > > Also thank you for raising your score even further post-rebuttal to a clear accept.
> > >
> > > Very much appreciated and kind regards,
> > > your authors

---

### Official Review · Reviewer_tWPL · 2023-07-06

**Soundness:** 3 good
**Presentation:** 2 fair
**Contribution:** 3 good
**Rating:** 6
**Confidence:** 3

**Summary:**

This work introduced a concept of consolidating causal mechanisms to transform large-scale Structural causal  models (SCMs) while preserving consistent interventional behaviour. The author shows consolidation is powerful for simplifying SCM, disscuss the complexity and give a perspective on generalization.

**Strengths:**

The author builds a solid framework of consolidating causal mechanisms. The expression of notations is adeque. The spirit of compression the causal equation is interesting and straight-forward.

**Weaknesses:**

1. The author should spend more efforts to describe the compression of causal equations clearly, as this is the key contribution of this work.
2. The author did not propose a concrete algorithm to compress the causal equation.

**Questions:**

1. Can author make a detail derivation on how to extract a consolidating causal mechanism from a toy SCM?
2. In what scale of SCMs can this method handle, and what is the complexity? Can author give some larger scale examples?

**Limitations:**

see the weakness and question parts.

---

> ### Author Rebuttal · Authors · 2023-08-09
>
> Thank you reviewer tWPL for detailing some important aspects that helped improve our paper. Here are some brief comments on what we've improved and the questions you've raised.
>
> * *Regarding discussing of compressibility*:
>
>   Compressing structural equations to a minimal representation is highly dependent on the equations under consideration and probably incomputable for most problems. As there is ultimately no way of measuring compressibility of equations by only considering their connecting graph structure, we opted to provide an discussion with regard to some of the information-theoretic implications that can be inferred in section 4.1. Thanks for highlighting this point. We have modified section 4.1 to more critically discuss this limitations of our approach.
>
> * *Regarding an algorithm for causal consolidation*:
>
>   Nonetheless, we've added the requested algorithm to the paper. Please find it attached PDF in the global response that shows the algorithm with an additional toy SCM for a step-by-step worked-out demonstration at the end of this reply. The presented concrete example consolidates a collider structure (consider consolidation of $\mathbf{A}_1$) and a chain (consider $\mathbf{A}_2$ of the example) matching the contents of section 4.1.
>
>   Thanks for pointing out both points, we consider especially the addition of the algorithm as a strong contribution to our paper.
>
> * *Regarding larger SCMs*:
>
>   In our example for the CoinRunner game where we consolidate agent behavior captured previously in a large graph as depicted in Figure 4, we can see that consolidation also works in larger structures. We've now added a paragraph to further discuss the general limitations one faces when performing consolidation on larger graphs. Thanks for pointing this out.
>
> We also look forward to more discussions with you.
>
>
> **An Example Application of the 'Consolidate' Algorithm**
>
> Consider the SCM provided in the general answer with its structural equations and resulting graph (endogenous variables are $B,C,D,E,F,G,H$ with only one exogenous $A$ with each structural equation highlighted on the r.h.s., note that the subscript on $f_x$ denotes the variable to be determined e.g.\ $B\leftarrow f_B(A)$).
>
> In the first step, the algorithm's user decides on a partition. Let's consider for instance the following partition i.e., allowed intervention and consolidation sets:
>
> $\mathcal{A} = \\{\\{E,F,G\\}, \\{B,C\\}, \\{D,H\\}\\}; \mathbf{E} = \\{ C, F, H \\};$
> $\mathcal{I} = \\{ \\{do(D = \text{true})\\}, \\{do(D = \text{false})\\}, \\{do(G = \text{false})\\} \\}$
>
> To finalize our example, a step-by-step application of \textsc{Consolidate} for the cluster $\mathbf{A}_1 = \{E,F,G\}$:
>
> *Step 3:* $\mathbf{E}_1 \gets \\{E,F,G\\} \cap \\{C, F, H\\} = \\{ F\\}$
>
> *Step 4:* $\mathbf{E}'_1 \gets \\{F\\} \cup (\text{pa}(\textbf{V} \setminus \\{ E,F,G \\}) \cap \\{ {E,F,G} \\}) = \\{F\\} \cup (\\{A,B,C,G\\} \cap \\{E,F,G\\}) = \\{F,G\\}$
>
> *Step 5:* $\textbf{U}_{\mathbf{A}_1} \gets \text{pa}(\\{E,F,G\\}) \setminus \\{E,F,G\\} = \\{A,E,F\\} \setminus \\{E,F,G\\} = \{A\}$
>
> *Step 6:* $\mathcal{I}_{\mathbf{A}_1} \gets \\{\\{ do(X_i = v) \in \mathbf{I}\ : X_i \in \\{ E,F,G\\}\\}: \mathbf{I} \in \mathcal{I}\\} = \\{\\{ do(G = \text{false}) \\}\\}$
>
> *Step 7:* $\rho_{\mathbf{E}'\_1} \gets \\{ f_E(A) := E \text{ mod } 5 = 0; f_F(A) := E \text{ mod } 10 = 0; f_G (E,G) := A \land B  \\}$
>
> *Step 8:* $\rho^\star_{\mathbf{E}'_1} \gets \text{argmin} \mathcal{K}(\rho\_{\mathbf{E}'\_1} ) = \\{\rho_F(A) := A\text{ mod } 10=0; \rho_G(\rho_F, \mathbf{I}\_{\mathbf{A}_1}) := \rho_F \land (do(G = \text{false}) \notin \mathbf{I}\_{\mathbf{A}_1})  \\}$
>
> *Step 9:* $\mathcal{M}\_{\mathbf{A}_1,\mathbf{E}} \gets (\\{F,G\\}, \\{F\\}, \rho^\star\_{\mathbf{E}'_1}, \\{\\{ do(G = \text{false}) \\}\\}, P_A)$
>
> Note how computing $f_E$ is no longer required. In a similar fashion, equations in $\mathbf{A}_2$ resemble a chain that can be composed: $f_C \circ f_B$ (previously called '\textit{stacked}'; cf. Sec. 4.1). Since $|\text{Img}(f_B)|=2$, at least one of the three conditions of $f_C$ (since $f_C$ is a 3-case function) will be discarded. (Eventually yielding $\rho^{\star}\_{\mathbf{E}'_2}{\gets}\\{ \rho_C(A){:=} A \leq 5\\}$). As $D$ is not in $\mathbf{E}$ and not required by any other sub SCM it can be marginalized. $A_3$ then reduces to $\rho^{\star}\_{\mathbf{E}'_3}{\gets}\\{ \rho_H(C,G) := C \lor G\\}$.

---

> > ### Author Response · Authors · 2023-08-16
> > **Looking Forward to Feedback on Our Response**
> >
> > Dear Reviewer,
> >
> > We appreciate the time and effort that you have taken to provide us with the review. We would like to ask if the reviewer has any further concerns or is satisfied by our responses to the original review.
> >
> > We are looking forward to any further discussion with the reviewer and would like to thank the reviewer again for helping make our paper better.
> >
> > Regards,
> >
> > The Authors

---

> > ### Comment · Reviewer_tWPL · 2023-08-21
> >
> > Thanks very much for the clarification. I acknowledge that I have read the rebuttal. I will keep my score.

---

### Official Review · Reviewer_CKKq · 2023-07-07

**Soundness:** 3 good
**Presentation:** 2 fair
**Contribution:** 2 fair
**Rating:** 6
**Confidence:** 3

**Summary:**

An operation of consolidation on SCMs is defined and its merits laid out in numerous examples. The operation amalgamates variables while preserving aspects of the causal structure. As opposed to the similar operation of marginalization that comes from probability theory and is well-known in causal abstraction, the consolidation operation is capable of representing interventions on those variables that were abstracted since the consolidated variable is of a special kind. It also accommodates compressions since other compatible functions can be assigned to the consolidated variable; this also opens the possibility of widening the domains of the original variables.

**Strengths:**

The problem is relatively well-situated within the literature, I believe the results to be sound, and the examples are good.

**Weaknesses:**

I am not convinced of the utility of the approach. What open problems does this contribute to solving?

**Questions:**

- The title seems inappropriate (especially imperative mood). Are you suggesting that consolidation is always superior to marginalization?
- Line 4: "Thus, ... analyze." is not a full sentence
- Lines 35--36: "Given that ... outcome." is not a full sentence. Awkward transition follows in "That is, ..."
- Line 48: "intervention preserving" -> "intervention-preserving"
- Line 69: it is inappropriate to cite do-calculus specifically here, rather, this is talking about the foundations of the SCM framework as a whole.
- Line 116: "Section 4 four" -> "Section 4"
- Is it really necessary to include the causal graph in the right of Figure 4?

**Limitations:**

Yes, they have.

---

> ### Author Rebuttal · Authors · 2023-08-09
>
> Thank you reviewer CKKq for your detailed review and appreciating our work and good examples. Your comments have helped us improve the paper. We hope to answer all points you made in the following:
>
> * *Regarding the title and its implications of "superiority" of consolidation to marginalization:*
>
>   The short answer is: Yes. The more involved answer is: all authors have discussed this thoroughly in advance and settled for the present title suggesting that we would not claim consolidation to "always [be] superior" due to its normative implications. Consolidation generalizes marginalization in the sense that marginalization can be modeled by consolidating with $\mathcal{I} = \emptyset$. Therefore, if we consider questions that involve causality / causal models, then consolidation necessarily needs to be considered since it can actually handle interventions (all cases where $\mathcal{I} \neq \emptyset$). Thus, if causal questions are not of concern, then marginalization can be considered as remaining the standard procedure. On another note we'd like to point out that marginalization does not require minimality of the resulting representation. Thank you for pointing this out since we are able to see how this is not immediately obvious from the paper and now we've added a corresponding statement to the refined version.
>
> * *Regarding Line 69 regarding the do-calculus citation:*
>
>   We fully agree and have corrected it in the paper.
>
> * *Regarding the necessity of the causal graph in the right of Figure 4:*
>
>   Our intention here was to showcase an example of consolidation for larger graphs. We think that having the graph included in Figure 4 makes it more clear that even domain experts in a certain area (here the specific CoinRunner game) might struggle to reduce the graph to extract (humanly) interpretable / meaningful descriptions of the agent behavior. Consolidation proved to work for this example as efficient as with some more elementary examples. Also the graph in Fig.4 helps to emphasize how consolidation can reduce overall complexity. Nonetheless, to give all the details we had added the full graph (fully readable) to the appendix as mentioned in the paper.
>
> * *Regarding the utility of consolidation and open problems that might be tackled:*
>
>   Thank you for pointing this out since this is a key aspect to why we think consolidation is important. In the following our perspective on this matter: as with marginalization, the operation of consolidation can help with simplifying causal models and therefore remove / abstract away variables that are irrelevant to the users' specific analysis. To include the effects of interventions (which is arguably the key operation in Pearl's causality framework and in more general, philosophical accounts of interventionist causality), the operations of marginalization and causal effect estimation would need to be repeated for every mutilated graph. With consolidation however we can retain the effects of interventions. As an example recall the domino example from Figure 1 / Appendix: we do not have to apply every possible intervention and summarize their outcomes, but can simply read from the consolidated formula, that every possible intervention will prevent the effect from arriving at the last domino. While the domino example specifically is not interesting for any particular analysis, it serves as a conceptual argument that highlights the implications to more important examples such as more classical causal analysis or our second example with agents interacting with an environment (robotics scenarios). More broadly speaking we intend for future work to follow up with researching applications of consolidation after having set the foundations of the operation in this work. Consolidation lays the foundation for justifying interventions onto high-level objects (modeled by consolidated SCM) where interventions can be applied externally but the inner structure of $\rho$ might not be known. Thanks again for highlighting this, we've added a discussion of this to the new paper version.
>
> Also thank you for pointing out various grammar and spelling mistakes. We have corrected all of them now.
>
> We look forward to any further discussion with you, thanks again.

---

> > ### Comment · Reviewer_CKKq · 2023-08-14
> >
> > I have read the response and will maintain my current rating.

---

> > > ### Author Response · Authors · 2023-08-15
> > >
> > > Thank you for both confirming that you've read our response and that you remain with your positive score.
> > > Thanks again for helping improving our paper through your feedback.
> > >
> > > If there is anything else, then please do let us know.
> > >
> > > Kind regards,
> > > your authors

---

### Author Rebuttal · Authors · 2023-08-09

Thanks again to all reviewers for thorough checking and commenting on our work, helping us improve it to its current form. In addition to the individual comments, please see attached the PDF with the generally requested consolidation algorithm. The algorithm summarizes the construction of causal compositional variables and partitioned SCM - as described in the paper - which we have added to the paper as a separate subsection right before the applied examples (sec. 4.2). We have also included a more generic worked-out example show-casing the actual steps of the algorithm and featuring the consolidation of a collider and a chain. Since NeurIPS this year restricts the PDF to only containing tables and figures we provided the actual worked out example in the answer of reviewer tWPL who explicitly requested this addition.

We look forward to any further discussion with you.

Kindly, your authors.

---

### Decision · Program_Chairs · 2023-09-21

**Decision:**

Accept (poster)

**Comment:**

The paper introduces and formalizes a process leading from a causal graph to an abstraction thereof, called "consolidation", meant to support interventions.

The contribution of the paper mostly is to lay the foundations for defining the consolidation operation; the discussion with the reviewers concluded to the relevance of the approach to operationalize large structural causal models.